🔓 | **Open Peer Review** | Mycology | Research Article

# The *Neosartorya (Aspergillus) fischeri* antifungal protein NFAP2 has low potential to trigger resistance development in *Candida albicans in vitro*

Gábor Bende,[1,2] Nóra Zsindely,[3] Krisztián Laczi,[1,4] Zsolt Kristóffy,[1] Csaba Papp,[5] Attila Farkas,[4] Liliána Tóth,[1] Szabolcs Sáringer,[6] László Bodai,[3] Gábor Rákhely,[1,7] Florentine Marx,[8] László Galgóczy[1,9]

**ABSTRACT**    Due to the increase in the number of drug-resistant *Candida albicans* strains, new antifungal compounds with limited potential for the development of resistance are urgently needed. NFAP2, an antifungal protein (AFP) secreted by *Neosartorya* (*Aspergillus*) *fischeri*, is a promising candidate. We investigated the ability of *C. albicans* to develop resistance to NFAP2 in a microevolution experiment compared with generic fluconazole (FLC). *C. albicans* adapted to only 1× minimum inhibitory concentration (MIC) of NFAP2, which can be considered tolerance rather than resistance, compared with 32× MIC of FLC. Genome analysis revealed non-silent mutations in only two genes in NFAP2-tolerant strains and in several genes in FLC-resistant strains. Tolerance development to NFAP2 did not influence cell morphology. The susceptibility of NFAP2-tolerant strains did not change to FLC, amphotericin B, micafungin, and terbinafine. These strains did not show altered susceptibility to AFPs from *Penicillium chrysogenum*, except one which had less susceptibility to *Penicillium chrysogenum* antifungal protein B. FLC-resistant strains had decreased susceptibility to terbinafine and NFAP2, but not to other drugs and AFPs from *P. chrysogenum*. NFAP2-tolerant and FLC-resistant strains showed decreased and increased NFAP2 binding and uptake, respectively. The development of tolerance to NFAP2 decreased tolerance to cell wall, heat, and UV stresses. The development of FLC resistance increased tolerance to cell wall stress and decreased tolerance to heat and UV stresses. Tolerance to NFAP2 did not have significant metabolic fitness cost and could not increase virulence, compared with resistance to FLC.

**IMPORTANCE**    Due to the increasing number of (multi)drug-resistant strains, only a few effective antifungal drugs are available to treat infections caused by opportunistic *Candida* species. Therefore, the incidence of hard-to-treat candidiasis has increased dramatically in the past decade, and the demand to identify antifungal compounds with minimal potential to trigger resistance is substantial. The features of NFAP2 make it a promising candidate for the topical treatment of *Candida* infection. Data on the development of resistance to antifungal proteins in *Candida albicans* are lacking. In this study, we provide evidence that NFAP2 has a low potential to trigger resistance in *C. albicans in vitro*, and the developed tolerance to NFAP2 is not associated with severe phenotypic changes compared with development of resistance to generic fluconazole. These results suggest the slow emergence of NFAP2-resistant *Candida* strains, and NFAP2 can reliably be used long-term in the clinic.

**KEYWORDS**    antifungal protein, fluconazole, *Candida albicans*, *Neosartorya* (*Aspergillus*) *fischeri*, resistance

**Peer Reviewer** Norman van Rhijn, The University of Manchester, Manchester, United Kingdom

Address correspondence to László Galgóczy, galgoczi@bio.u-szeged.hu.

The authors declare no conflict of interest.

See the funding table on p. 19.

In the last two decades, the occurrence of opportunistic fungal infections (FIs) has increased due to the rise in the number of patients with immunosuppression and resistance to antifungal drugs. This increase poses a serious threat to public health globally (1). In addition to causing nonlife-threatening cutaneous mycoses in immunocompetent individuals, fungi cause subcutaneous or systemic and disseminated fatal infections in patients with a weak and impaired immune system due to HIV infection, hematological malignancies, diabetes mellitus, immunosuppressive therapy, or intensive care of organ transplantation (2, 3). FIs have generated little public attention compared with tuberculosis and malaria, although the number of annual deaths (~1.7 million) due to FIs is close to the number of deaths due to tuberculosis and malaria together (4). Therefore, the World Health Organization released the first fungal priority pathogens list in October 2022, which highlights 19 fungal pathogens that pose the most serious threat to human health. Four of the listed fungi belong to the genus *Candida*, while two were formerly classified in this genus (5).

Candidiasis, caused by *Candida* species, is the most common FI (6) and represents one of the most prevalent hospital-acquired infections (7). *Candida albicans* is responsible for most cases; however, an epidemiological shift to non-*albicans Candida* (NAC) species, represented by *Nakaseomyces glabrata* (formerly *Candida glabrata*), *Candida tropicalis, Candida parapsilosis*, and *Pichia kudriavzevii* (formerly *Candida krusei*), has been observed recently. Depending on the age group and geographical location of the patient, the NAC species can exceed *C. albicans* (8). The emergence of multidrug-resistant nosocomial *Candida auris* is a cause of alarm in epidemiology (2). Treatment of *Candida* infection is based on a relatively shallow pool of four antifungal drug classes: azoles, echinocandins, polyenes, and fluorinated pyrimidines (9). Fluconazole (FLC) is the most commonly used fungistatic agent for treating candidiasis; therefore, it is not surprising that FLC resistance is common among human pathogenic *Candida* species (10). Resistance against a certain class of antifungal drugs severely hampers good therapeutic outcome and requires the administration of other classes of antifungals (1). The increased use of another class of antifungals in combination with a less effective drug increases the risk of multidrug resistance (9). For example, the introduction of echinocandins to treat *N. glabrata* infections clinically led to the development of multidrug resistance to azoles–echinocandins (11). Emerging (multi)drug-resistance and lack of a new class of antifungal agents with novel mechanisms of action have led to an urgent demand for new antifungal strategies to treat FIs caused not only by *Candida* species but also other human fungal pathogens.

Several new antifungal drugs are in various stages of clinical trials. They offer new avenues for the treatment of FIs due to their novel modes of action, with reduced toxicity, less adverse effects, and minimal resistance development potential compared with conventional drugs. Fosmanogepix (APX001), ibrexafungerp (SCY-078), and VT-1161 are among the most promising phase III clinical trial candidates, which are highly active against several *Candida* species, including conventional drug-resistant *C. albicans* and *C. auris* isolates (12). Although several new antifungal compounds are in the early stages of clinical trial, very few have reached the later stages, which do not guarantee marketed treatment (13). Therefore, the search for new antifungal compounds is essential. Biomolecules with antifungal effects are promising alternatives (14).

Antifungal proteins (AFPs) are promising drug candidates for treating mycoses because their mode of action differs from conventional antifungal medicines. AFPs can provide less toxic treatment options compared with licensed antifungals, which cause serious adverse effects after prolonged therapy (15). AFPs of filamentous fungal origin effectively inhibit the growth of different *Candida* species *in vitro* (16). NFAP2, an AFP derived from the ascomycete *Neosartorya* (*Aspergillus*) *fischeri* NRRL 181, has high therapeutic potential in the treatment of superficial *Candida* infections (17, 18). This small molecular weight (5.6 kDa), cationic, and highly stable extracellular protein is remarkably active against not only planktonic cells of different *Candida* spp. (19) but also sessile biofilm cells of fluconazole-resistant *C. albicans* and multidrug-resistant *C. auris*

isolates (17, 20). NFAP2 can interact synergistically with azoles, echinocandins (first-line antifungal drugs for superficial and/or invasive *Candida* infection), and amphotericin B (AMB), significantly decreasing their minimum inhibitory concentrations (MIC) *in vitro* (17, 19, 20). In a murine vulvovaginal candidiasis model, NFAP2 demonstrated its monotherapeutic potential against an FLC-resistant strain of *C. albicans*. Coadministration of NFAP2 and FLC during therapy resulted in synergy *in vivo* and reversed resistance; moreover, none of the treatments caused morphological alterations and serious pathological reactions in vaginal and vulvar tissues (17). Consistently, in a three-dimensional human cell-based skin infection model, NFAP2 was a safe and effective topical agent. NFAP2 significantly decreased the fungal burden of *C. albicans* in the stratum corneum and was well-tolerated by the model because alterations to an intact outside-in permeability barrier were not observed after NFAP2 treatment (18).

Therefore, NFAP2 is a promising therapeutic agent for treating superficial drug-resistant *Candida* infections. One of the requirements for the reliable therapeutic application of NFAP2 involves understanding the potential for development of NFAP2 resistance in fungi.

In this study, our objective was to investigate the potential of NFAP2 to trigger resistance development in *C. albicans* compared with FLC, a widely used anti-*Candida* drug, which is known to commonly induce resistance in *Candida* (10). The results of the microevolution experiments showed that *C. albicans* has limited ability to develop strong resistance to NFAP2, exhibiting merely tolerance, compared with FLC. The development of tolerance to NFAP2 does not have a fitness cost and does not substantially alter cell morphology and susceptibility to conventional antifungal drugs, except for FLC. However, it affects the uptake of NFAP2, tolerance to abiotic stress, and virulence of *C. albicans*.

## RESULTS

### Generation of NFAP2- and FLC-resistant *C. albicans* strains

Microevolution experiments, including six-six independent parallels, were performed to investigate the adaptation potential of *C. albicans* to increasing concentrations of NFAP2, compared with FLC, a widely used anti-*Candida* drug (21). Other microevolution studies have shown that *C. albicans* can adapt to increasing FLC concentration and evolve strong resistance (22). Therefore, FLC served as a control for the development of adaptation and resistance mechanisms in our study. First, the susceptibilities of *C. albicans* CBS 5982 to NFAP2 and FLC were determined in a low-cation medium (LCM) for the microevolution experiments. The MICs of NFAP2 and FLC were 3.125 and 128 µg mL$^{-1}$, respectively. Adaptation analysis in the microevolution experiment showed that *C. albicans* CBS 5982 could adapt only to 1× MIC of NFAP2 (3.125 µg mL$^{-1}$) and did not survive serial passage in the presence of 2× MIC of NFAP2 (Fig. 1A). By contrast, *C. albicans* CBS 5982 could grow even at 32× MIC of FLC (4, 096 µg mL$^{-1}$; Fig. 1B). These observations indicated that *C. albicans* has lower potential to adapt to NFAP2 than FLC; furthermore, adaptation to merely 1× MIC of NFAP2 suggests that this phenomenon is more indicative of tolerance than resistance. The cultures from the adaptation experiment underwent serial passages in antifungal-free medium and cultured on selective LCM agar plates containing 1× MIC of NFAP2 or 32× MIC of FLC to remove strains that do not have acquired tolerance or resistance. This last step of the microevolution experiment was selected for stably tolerant or resistant strains of *C. albicans* that evolved tolerance to 1× MIC of NFAP2 or resistance to 32× MIC of FLC. Three colonies each of independently evolved *C. albicans* cultures tolerant to NFAP2 and resistant to FLC were randomly selected. These acquired strains were named NFAP2/4, NFAP2/5, and NFAP2/6 (NFAP2-tolerant), as well as FLC/3, FLC/5, FLC/6 (FLC-resistant) strains, respectively. Results of the microevolution experiment showed that *C. albicans* has limited ability to develop pronounced resistance to NFAP2, exhibiting only tolerance, compared with the conventional triazole antifungal FLC.

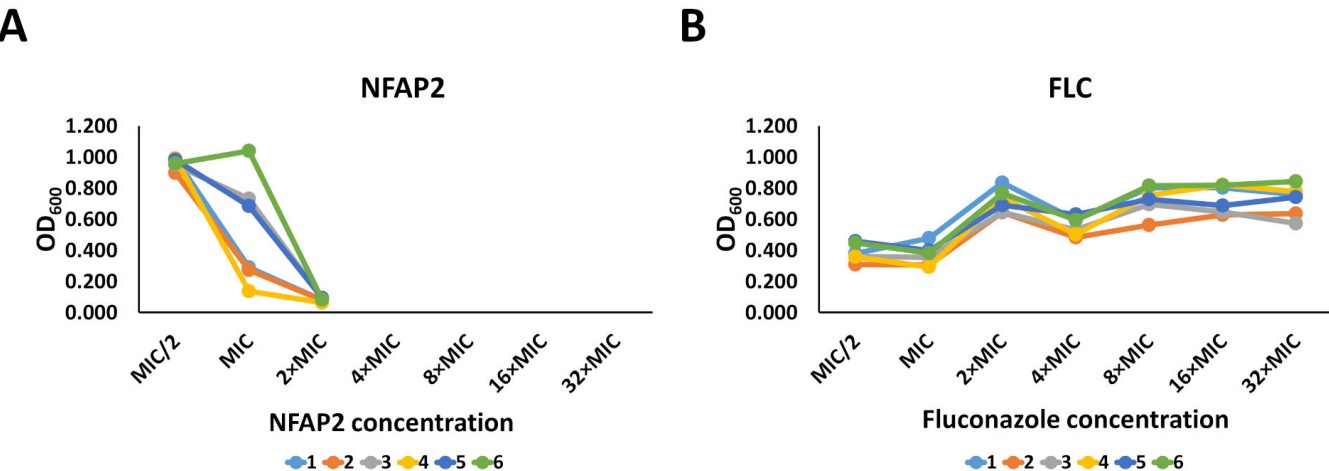

**FIG 1** Adaptation of *C. albicans* CBS 5982 to increasing concentrations of NFAP2 (A) and FLC (B) in a microevolution experiment performed in an LCM. The optical densities of cultures of six independent lineages (1–6) were measured at the final passage of each concentration. The MICs of NFAP2 and FLC were 3.125 and 128 µg mL$^{-1}$, respectively.

## Whole-genome sequencing of NFAP2-tolerant and FLC-resistant *C. albicans* strains

The whole genomes of wild-type *C. albicans* CBS 5982 (WT), three independently evolved NFAP2-tolerant and FLC-resistant strains were sequenced to reveal genomic changes under NFAP2 or FLC pressure and to identify mutations in the tolerant and resistant lines. Three strains of non-selected control lines that had not been subjected to drug treatment were also sequenced and compared to exclude the influence of the experimental condition on the genome mutation. The assembled CBS 5982 WT genome has 94.3% estimated genome completeness and 98.9% average nucleotide identity with 72.4% coverage compared to the SC5314 reference genome. GlimmerHMM (23) was used for gene prediction, which identified 2,983 genes. Considering that *C. albicans* genome consists of ~6,100 genes (24), we performed an additional gene prediction using Funannotate (25), which identified 5,951 genes. These gene predictions for the WT genome were used as a reference during variant analysis to detect gene mutations in the genomes of the other strains. Considering that *C. albicans* is a diploid yeast, sequences of mutation-affected regions of these genes were checked for heterozygosity and compared with WT sequences to ensure that these are not artifacts of the genome analysis software. Figure S1 shows the sequenograms of the verified mutant gene regions compared with the WT. Whole-genome sequencing (WGS) revealed a deletion in a biofilm-induced uncharacterized protein (UBIP; National Center for Biotechnology Information [NCBI] accession number: XM_709449.2, GenBank: KHC66060.1) in NFAP2/4. NFAP2/5 and NFAP2/6 carried the same multiple nucleotide variation (MNV) at the same position in a putative glycosylphosphatidylinositol-anchored protein PGA58 (NCBI accession number: XM_715546.1). For PGA58, the presence of the heterozygous mutation could not be verified because we obtained noisy and incomplete spectra even after several repetitions of the PCR with different primer pairs (data not shown). FLC/3 contained two single nucleotide variations (SNVs) and one nucleotide insertion in clathrin heavy chain 1 (CHC1; NCBI accession number: XM_705744.2) and two nucleotide deletions in serine/threonine-specific protein phosphatase of type 2C-related family (PTC2; NCBI accession number: XM_713113.2). FLC/5 had one SNV in phosphoinositide 5-phosphatase (INP51; NCBI accession number: XM_709454.2). FLC/6 contained one SNV in an alkaline-responsive transcriptional regulator RIM101 (NCBI accession number: XM_709954.1). These results are summarized in Table 1. The locations of the mutations in the translated protein sequences are presented in Table S1 and Fig. S2.

**TABLE 1** NCBI BLAST identification of mutated genes in *C. albicans* CBS 5982 after NFAP2 or FLC pressure in a microevolution experiment[a]

| Strain | Gene match NCBI BLAST | NCBI accession | Identity (%) | Mutation | Base change | Amino acid change | WT reference (5'–3') | Mutant allele (5'–3') | Mutation zygosity |
|---|---|---|---|---|---|---|---|---|---|
| NFAP2/4 | UBIP | XM_709449.2 | 98.29 | Deletion | TCG → Ø | Ser145 → Ø | CAATTT[**TCG**]CCTCCT | CAATTT[**Ø**]CCTCCT | Heterozygous |
| NFAP2/5 | PGA58 | XM_715546.1 | 93.1 | MNV | TG → CA | Val201 → Ala | TCTGGAG[**TG**]GCTGGG | TCTGGAG[**CA**]GCTGGG | Heterozygous |
| NFAP2/6 | | | | | | | | | |
| FLC/3 | CHC1 | XM_705744.2 | 99.52 | SNV | G → A | Val480 → Ile | ACTTGCT[**G**]TTTATAT | ACTTGCT[**A**]TTTATAT | Heterozygous |
| | | | | | C → A | Ala479 → Asp | GCACTTG[**C**]TGTTTAT | GCACTTG[**A**]TGTTTAT | |
| | | | | Insertion | Ø → T | Ala479 → STOP | TGGCAC[**TT**]GCTGTTT | TGGCAC[**TTT**]GCTGTTT | |
| | PTC2 | XM_713113.2 | 99.74 | Deletion | G → Ø | Glu438 → STOP | AGAAGAA[**G**]AAGAAGG | AGAAGAA[**Ø**]AAGAAGG | |
| | | | | | GAAGA → Ø | Glu436 → STOP | AGAA[**GAAGA**]AGAAG | AAGAA[**Ø**]AGAAG | |
| FLC/5 | INP51 | XM_709454.2 | 99.33 | SNV | G → T | Gly899 → Val | GGTCTTG[**G**]TGGAGTA | GGTCTTG[**T**]TGGAGTA | Heterozygous |
| FLC/6 | RIM101 | XM_709954.1 | 98.99 | SNV | A → G | Tyr576 → Cys | GACGATT[**A**]CAGCACT | GACGATT[**G**]CAGCACT | Heterozygous |

[a]Base changes are indicated with bold letters and are in square brackets. The detailed datasheet file of variant analysis can be found in Supplementary File 1 and 2. CHC1: clathrin heavy chain 1, FLC/3, 5, 6: FLC-resistant strains, INP51: phosphoinositide 5-phosphatase, NFAP2/4, 5, 6: NFAP2-tolerant strains, PGA58: putative glycosylphosphatidylinositol-anchored protein, PTC2: serine/threonine-specific protein phosphatase, RIM101: alkaline-responsive transcriptional regulator, STOP: termination codon, UBIP: uncharacterized biofilm-induced protein, WT: wild-type, Ø: nucleotide(s) not present.

## Investigation of cell morphology

Cell morphologies of WT, NFAP2-tolerant, and FLC-resistant strains were investigated with scanning electron microscopy to examine whether genomic changes led to morphological alterations (Fig. 2A in 15,000× magnification and Fig. S3A in 2,000× magnification). Cells of all strains were round-to-oval and secreted extracellular polymeric substance (EPS). EPS secretion, however, was less in NFAP2/6, FLC/5, and FLC/6, compared with the WT. FLC/3 exhibited a wrinkled surface, while the WT and the other strains showed a smooth cell surface (Fig. 2A). After NFAP2-treatment (Fig. 2B in 15,000× magnification and Fig. S3B in 2,000× magnification), WT, NFAP2/4, NFAP2/5, FLC/3, and FLC/5 did not secrete EPS (Fig. 2B; Fig. S3B), and some WT cells collapsed and lost their ovoid shape (Fig. S3B). Thus, it seems that the development of tolerance to NFAP2 does not influence cell morphology.

## Antifungal susceptibility testing to conventional antifungal drugs and antifungal proteins

Development of resistance to an antifungal drug usually results in cross-resistance to other drugs belonging to different antifungal classes (26). Antifungal susceptibility tests were performed in LCM to investigate whether the developed resistance influences the susceptibility of *C. albicans* to conventional antifungal agents and ascomycetous anti-*Candida* AFPs. We tested antifungals including the polyene AMB, triazole FLC, echinocandin micafungin (MFG), allylamine terbinafine (TRB), and NFAP2, as well as AFPs from *Penicillium chrysogenum*, *viz.* PAF, PAFγ$^{opt}$ (27), PAFB (28), and PAFC (29). Susceptibilities of the WT strain to these drugs and AFPs were considered 1× MIC. Susceptibility tests indicated that the developed tolerance or resistance did not influence the susceptibilities of NFAP2-tolerant and FLC-resistant strains to AMB and MFG and the susceptibility of NFAP2-tolerant strains to terbinafine (TRB). Interestingly, the susceptibility of FLC-resistant strains decreased to 2× MIC of TRB. Regarding FLC, the susceptibilities of NFAP2-tolerant strains increased to 0.5× MIC, and the susceptibilities of FLC-resistant strains decreased to 32× MIC. PAF and PAFC were ineffective even at the highest investigated concentration (50 µg mL$^{-1}$). The susceptibilities of all tolerant or resistant strains to PAFγ$^{opt}$ and PAFB were unchanged, except for NFAP2/6, which showed reduced susceptibility to PAFB (2× MIC). Susceptibility data are summarized in Table 2. Taken together, the susceptibility data indicated that development of tolerance to NFAP2 does not influence or increase susceptibility to conventional antifungal drugs, except FLC, and can decrease susceptibility to other AFPs. By contrast, FLC resistance can decrease susceptibility to TRB.

## NFAP2 uptake analysis

Studies have suggested that NFAP2 mediates its antifungal activity by disrupting the cell membrane of *C. albicans* (17, 19), which requires NFAP2 to bind to the cell. Therefore, we investigated whether reduced NFAP2 binding to the outer cell layers is responsible for a decreased susceptibility to NFAP2 in NFAP2-tolerant and FLC-resistant *C. albicans* strains. Therefore, NFAP2 was labeled with the green fluorophore boron-dipyrromethene (BODIPY; Bd-NFAP2) to follow its interaction with the yeast cell. Confocal laser scanning microscopy (CLSM) and fluorescence-activated cell sorting (FACS) analyses were conducted to follow and monitor NFAP2 binding and uptake in *C. albicans*. *C. albicans* cells were co-stained with propidium-iodide (PI) to detect cell death. CLSM analysis showed attachment of Bd-NFAP2 to the outer layers of WT and FLC-resistant *C. albicans* cells in 30 min, followed by accumulation of Bd-NFAP2 in this region or intracellularly, resulting in cell death within 4 h (Fig. S4). By contrast, attachment of Bd-NFAP2 to the outer cell layers and/or its internalization was not observed or not prominent in NFAP2-tolerant cells during this period (Fig. S4). To quantify *C. albicans* cells that had been treated with Bd-NFAP2 and interacted with the antifungal protein, FACS analysis was performed. After 16 h of incubation, FACS results showed a significant decrease

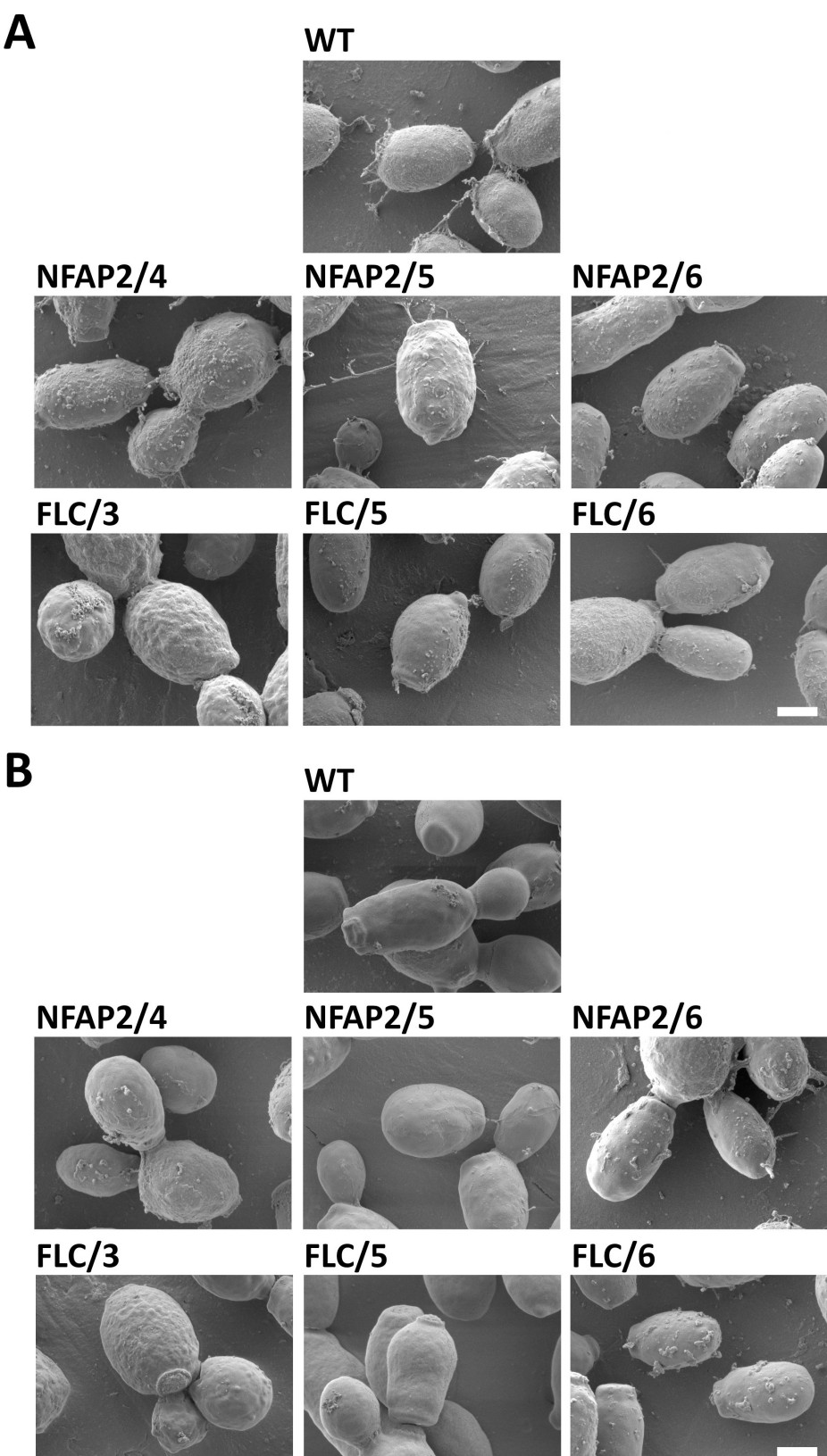

**FIG 2** Scanning electron microscopy of the parental wild-type *C. albicans* CBS 5982 (WT), NFAP2-tolerant (NFAP2/4, 5, 6), and FLC-resistant (FLC/3, 5, 6) strains (A), and the impact of NFAP2-treatment (1× MIC, 30 min, 30°C, 160 rpm, LCM) on cell morphology (B). 15,000× magnification. The scale bars represent 1 µm.

**TABLE 2** MIC of conventional antifungal drugs and ascomycetous AFPs in LCM[a]

| Strain | MIC (µg mL$^{-1}$) | | | | | | | | |
|---|---|---|---|---|---|---|---|---|---|
| | AMB | FLC | MFG | TRB | NFAP2 | PAF | PAFγ$^{opt}$ | PAFB | PAFC |
| WT | 1 | 128 | 0.125 | 64 | 3.125 | >50 | 50 | 25 | >50 |
| NFAP2/4 | 1 | 64 | 0.125 | 64 | 6.25 | >50 | 50 | 25 | >50 |
| NFAP2/5 | 1 | 64 | 0.125 | 64 | 6.25 | >50 | 50 | 25 | >50 |
| NFAP2/6 | 1 | 64 | 0.125 | 64 | 6.25 | >50 | 50 | 50 | >50 |
| FLC/3 | 1 | 8,192 | 0.125 | 128 | 6.25 | >50 | 50 | 25 | >50 |
| FLC/5 | 1 | 8,192 | 0.125 | 128 | 6.25 | >50 | 50 | 25 | >50 |
| FLC/6 | 1 | 8,192 | 0.125 | 128 | 6.25 | >50 | 50 | 25 | >50 |

[a]Susceptibilities of the WT strain to antifungal drugs and AFPs were considered 1× MIC. FLC/3, 5, 6: FLC-resistant strains, NFAP2/4, 5, 6: *Neosartorya* (*Aspergillus*) *fischeri* antifungal protein 2-tolerant strains, PAF: *Penicillium chrysogenum* antifungal protein, PAFγ$^{opt}$: γ-core optimized PAF, PAFB: *Penicillium chrysogenum* antifungal protein B, PAFC: *Penicillium chrysogenum* antifungal protein C, WT: wild-type *C. albicans* CBS 5982.

in Bd-NFAP2 interaction with each NFAP2-tolerant strain (NFAP2/4: 9.26% ± 2.51%, NFAP2/5: 18.39% ± 8.66%, NFAP2/6: 3.34% ± 2.96%) compared with the WT (23.45% ± 0.58%; Table 3). Bd-NFAP2 uptake was significantly increased in FLC/3 (45.51% ± 0.28%) and FLC/6 (31.59% ± 0.69%) and significantly decreased in FLC/5 (20.25% ± 1.08%). The proportions of dead cells after Bd-NFAP2 uptake are significantly decreased in NFAP2/5 (48.47% ± 4.01%), NFAP2/6 (54.43% ± 19.70%), FLC/5 (51.43% ± 2.33%), and FLC/6 (55.60% ± 1.49%), while increased in NFAP2/4 (76.71% ± 9.07%), and did not change significantly in FLC/3 (62.49% ± 10.04%) compared with the WT (63.03% ± 18.17%; Table 3). These results indicate that development of tolerance to NFAP2 and resistance to FLC can decrease and increase NFAP2 binding and uptake in *C. albicans*, respectively. FACS data further show that differences in the proportions of dead cell population in Bd-NFAP2-positive cells did not directly correlate with changes in binding/uptake rates between strains.

## Stress tolerance analyses

Triazole resistance can influence the stress response in *Candida* (26); therefore, to study whether the development of tolerance to NFAP2 influences response to abiotic stress, we conducted spot assays on LCM agar plates. The growth ability of the tolerant and resistant strains was evaluated in the presence of membrane stressors (NaCl and sodium dodecyl sulfate SDS) or cell wall stressors (calcofluor white [CFW]), heat treatment or UV irradiation compared with the WT. Growth ability of NFAP2/4, NFAP2/6, FLC/5, and FLC/6 was slightly decreased on this medium compared with the WT (Fig. 3A). All strains could tolerate the highest applied concentration of membrane stressors (300 mM NaCl and 100 µg mL$^{-1}$ SDS; Fig. 3B), and differences in their growing abilities were not observed. By contrast, NFAP2/4 and NFAP2/6 could not tolerate 10 µg mL$^{-1}$ CFW compared with other strains, whereas FLC/3 had better tolerance (Fig. 3B). Below this CFW concentration, tolerance of these two strains was not different from that of the WT (data not shown).

**TABLE 3** FACS analysis of Bd-NFAP2 uptake by wild-type, NFAP2-tolerant, and FLC-resistant *C. albicans* strains (%Bd-NFAP2$^+$) and the consequent cell death (%PI$^+$)[a]

| Strain | Bd-NFAP2+ % | Pi+ % |
|---|---|---|
| WT | 23.45 ± 0.58 | 63.03 ± 18.17 |
| NFAP2/4 | 9.26[b] ± 2.51 | 76.71[b] ± 9.07 |
| NFAP2/5 | 18.39[b] ± 8.66 | 48.47[b] ± 4.01 |
| NFAP2/6 | 3.34[b] ± 2.96 | 54.43[b] ± 19.70 |
| FLC/3 | 45.51[b] ± 0.28 | 62.49 ± 10.04 |
| FLC/5 | 20.25[b] ± 1.08 | 51.43[b] ± 2.33 |
| FLC/6 | 31.59[b] ± 0.69 | 55.60[b] ± 1.49 |

[a]Bd-NFAP2: BODIPY-labeled, Bd-NFAP2+: Proportion of Bd-NFAP2+ cells in the 30,000 cells detected, FLC/3, 5, 6: FLC-resistant strains, NFAP2/4, 5, 6: NFAP2-tolerant strains, PI+: Proportion of PI+ cells in Bd-NFAP2+ detected cells, WT: wild-type *C. albicans* CBS 5982.
[b]$P ≤ 0.05$ from type II ANOVA (analysis of variance ) followed by Bonferroni's correction as a post hoc test.

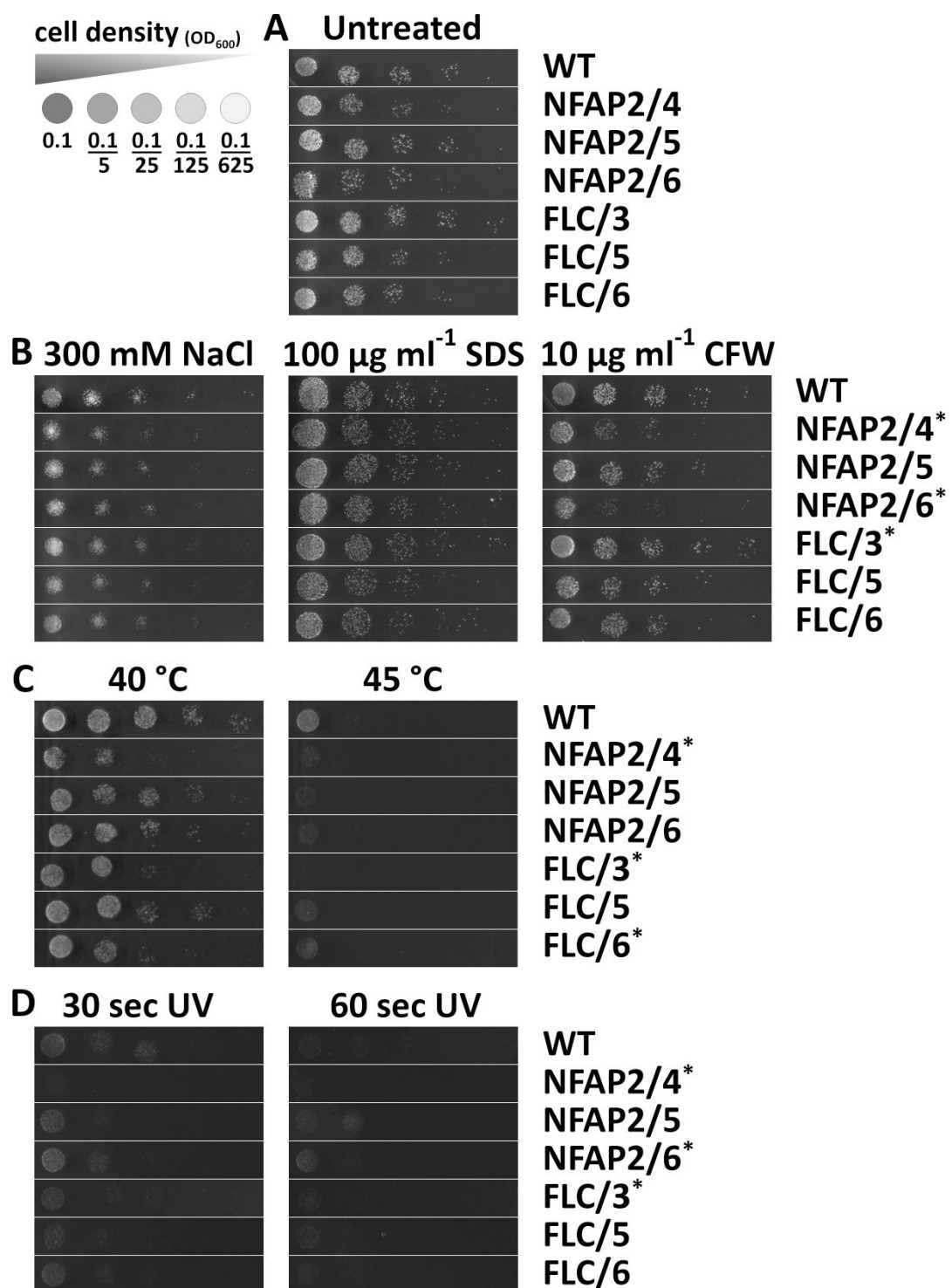

**FIG 3** Tolerance of *C. albicans* strains tolerant to NFAP2 (NFAP2/4, 5, 6) and resistant to FLC (FLC/3, 5, 6) to different abiotic stresses compared with the parental wild-type CBS 5892 (WT) strain. To test growth ability, starting from 0.1 $OD_{600}$ value, five-step 5× serial dilutions of the strains were spotted on LCM agar plates without supplementation (A), LCM with plasma membrane (NaCl) or cell wall stressors (B), after 30 min heat treatment at 40°C or 45°C (C), and after UV irradiation for 30 or 60 s (D). Gamma settings were adjusted to uniformize background tones for illustrative purposes. *: Change in the stress tolerance compared with WT.

Heat treatment at 40°C and 45°C remarkably decreased the growth of NFAP2/4, FLC/3, and FLC/6 compared with the WT (Fig. 3C). After heat treatment at 50°C, none of the strains grew on the plates. UV irradiation for 30 or 60 s decreased or fully inhibited

the growth of NFAP2/4, NFAP2/6, and FLC/3 (Fig. 3D). Two-minute-long UV irradiation killed all strains (data not shown). In summary, the development of resistance to NFAP2 can decrease tolerance to cell wall, heat, and UV stresses, whereas the development of resistance to FLC can increase tolerance to cell wall stress and decrease tolerance to heat and UV stresses.

## Metabolic fitness analysis

Development of FLC resistance can reduce the fitness of *C. albicans* in the absence of the drug (30); therefore, we investigated whether NFAP2 tolerance influences metabolic adaptation to different media compared with the WT. The growth abilities and kinetics of the NFAP2-tolerant and FLC-resistant strains were monitored in LCM, which was used for the microevolution experiments, malt extract (ME) as rich and Vogel's as minimal media, and Roswell Park Memorial Institute (RPMI)-1640 as standard clinical susceptibility test medium (31). Analysis of growth curves indicated that none of the tolerant and resistant strains had significantly different growing abilities in LCM compared with the WT. The growth of FLC/3 and FLC/5 significantly decreased in ME. In RPMI-1640, each FLC-resistant strain showed significantly weaker growth compared with the WT. In Vogel's medium, only FLC/5 exhibited reduced growth ability (Fig. 4). According to the growth curves, no significant differences were observed in the metabolic adaptation of NFAP2-tolerant strains in any medium compared with the WT (Fig. 4). These data indicated that tolerance to NFAP2 does not have a significant metabolic fitness cost compared with resistance to FLC.

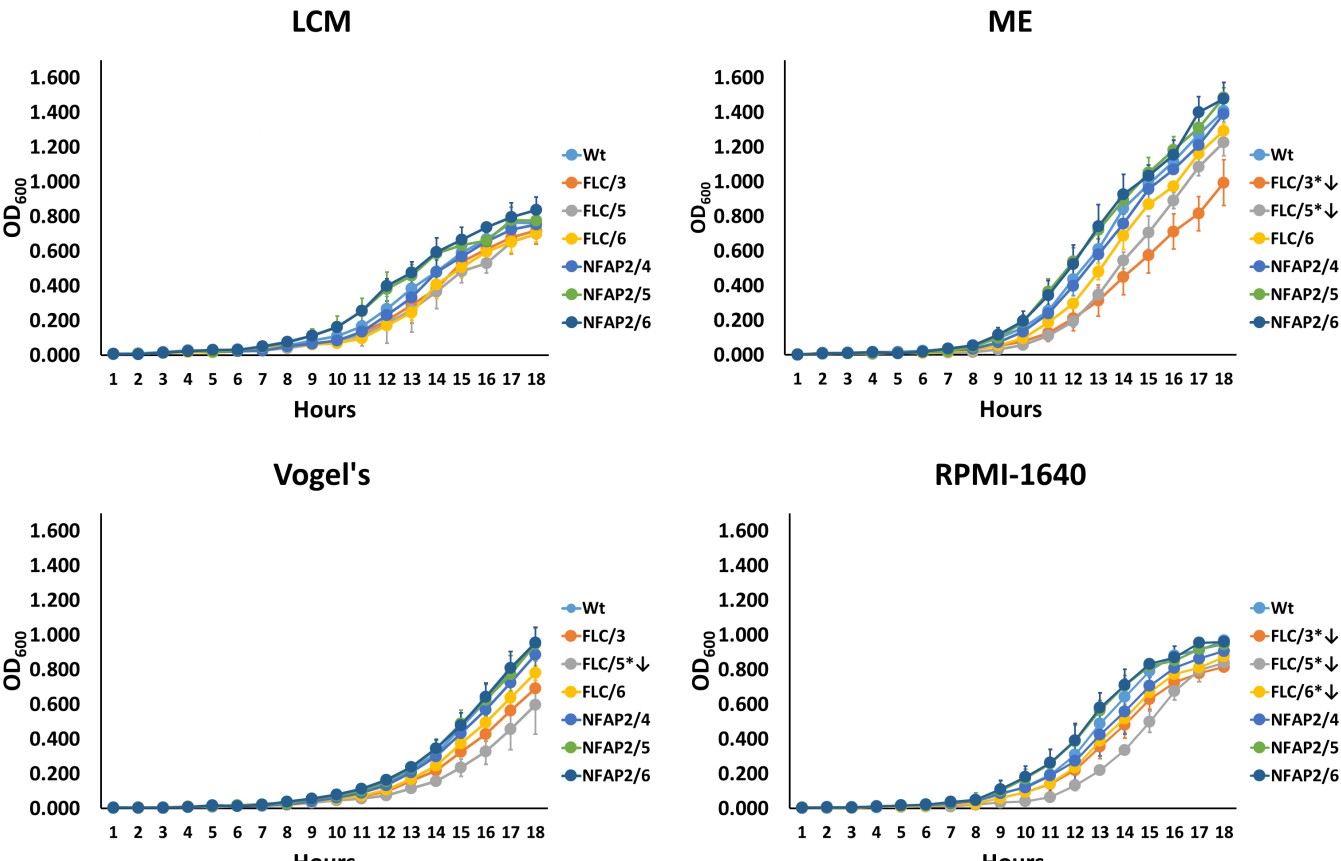

**FIG 4** Growth curves of *C. albicans* strains tolerant to NFAP2 (NFAP2/4, 5, 6) and resistant to FLC (FLC/3, 5, 6) in different media compared with the parental wild-type CBS 5982 (WT) strain (30°C, 160 rpm). Vogel's: Vogel's medium. *P ≤ 0.05 from repeated measure ANOVA followed by Tukey–Kramer's post hoc comparisons. ↓: significantly decreased growth.

## Virulence analysis

Development of resistance to FLC can reduce or increase the virulence of *Candida* species depending on the genomic background of the resistant phenotype (32). Therefore, we investigated the virulence of NFAP2-tolerant and FLC-resistant strains in a *Galleria mellonella* larval virulence model (33). Survival analysis of larvae infected with the tolerant and resistant *C. albicans* strains indicated that FLC resistance increased (FLC/5 and FLC/6) or decreased (FLC/3) virulence (Fig. 5B). By contrast, NFAP2 tolerance did not influence (NFAP2/4 and NFAP2/5) or decreased (NFAP2/6) virulence (Fig. 5A). Thus, compared with FLC resistance, NFAP2 tolerance does not enhance the virulence of *C. albicans*.

## DISCUSSION

AFPs of filamentous ascomycetes are potent agents for treating topical *Candida* infection (16). Several proof-of-concept studies support this hypothesis. These studies showed *in vitro* susceptibilities of various *Candida* spp. to different AFPs, interaction of AFPs with licensed antifungals, potential safe therapeutic applicability of AFPs, and their therapeutic efficacy in mucosal and cutaneous *Candida* infection models (16–18). Despite these promising results, studies on whether *Candida* spp. can develop resistance to fungal AFPs are lacking. These studies are essential to demonstrate the AFPs' long-term therapeutic applicability in the clinic, where resistance to conventional antifungal drugs is a challenge (1), and new antifungal compounds are needed. Therefore, we examined the potential for developing resistance to one of the most effective AFPs, NFAP2, against the most prevalent human pathogenic yeast, *C. albicans*, compared with generic FLC.

*C. albicans* has remarkable genome plasticity, which helps it overcome the harmful effects of selective environments (34, 35). Microevolutionary studies have described the high ability of *C. albicans* to develop strong resistance to FLC, as well as the genomic background of the observed FLC resistance in *C. albicans* (22, 36, 37). Similar data with AFPs are only available for *Nicotiana alata* defensin NaD1 (38) and a human salivary protein histatin 3 (39, 40) in *Saccharomyces cerevisiae* and *C. albicans*, respectively.

In our experimental setup of microevolution, *C. albicans* could grow even at 32× MIC of FLC, the highest applied drug concentration, compared with 1× MIC of NFAP2. This result indicates that *C. albicans* has less ability to develop strong resistance mechanisms to NFAP2, compared with FLC (Fig. 1). Consistently, *C. albicans* has been reported to develop resistance to >200× MIC of FLC (28, 41, 42), while *S. cerevisiae* only adapted to 10× MIC of NaD1 (38), and *C. albicans* became less susceptible to the cell killing effects of histatin 3 (39) in microevolution experiments. Resistance to an antifungal agent can lead to a change in susceptibility to antifungal drugs (38, 43–45). We observed this in the susceptibility of NFAP2-tolerant strains to FLC and FLC-resistant strains to TRB (Table 2). Development of resistance to FLC (22, 30, 37) and NaD1 (38) is associated with genomic changes that can influence cell morphology, metabolic fitness, stress tolerance, or virulence. We also observed changes in the cell morphology of FLC/3 (Fig. 2), metabolic fitness of FLC-resistant strains (Fig. 4), stress tolerance of NFAP2/4, NFAP2/6, FLC/3 strains (Fig. 3), virulence of NFAP2/6, and all FLC-resistant strains (Fig. 5A and B).

The investigation is essential whether the experimental setup of microevolution leads to mutations in functional genes, which could be linked to resistant phenotypes, and results in changes in antifungal susceptibility, metabolic fitness, stress tolerance, and virulence. Therefore, genome analysis of randomly selected FLC-resistant (FLC/3, FLC/5, and FLC/6) and NFAP2-tolerant (NFAP2/4, NFAP2/5, and NFAP2/6) strains was conducted and compared with the genome of the parental WT strain (*C. albicans* CBS 5982) and control lines that had not been treated with the antifungal compounds. Variant analysis showed heterozygous gene mutations in the genome of all tolerant and resistant strains (Table 1). Considering that *C. albicans* is a diploid yeast, heterozygous mutations can alter cell behavior (46) and modulate drug resistance (47). This phenomenon can explain the observed changes in antifungal susceptibilities of the evolved strains in our study (Table 2).

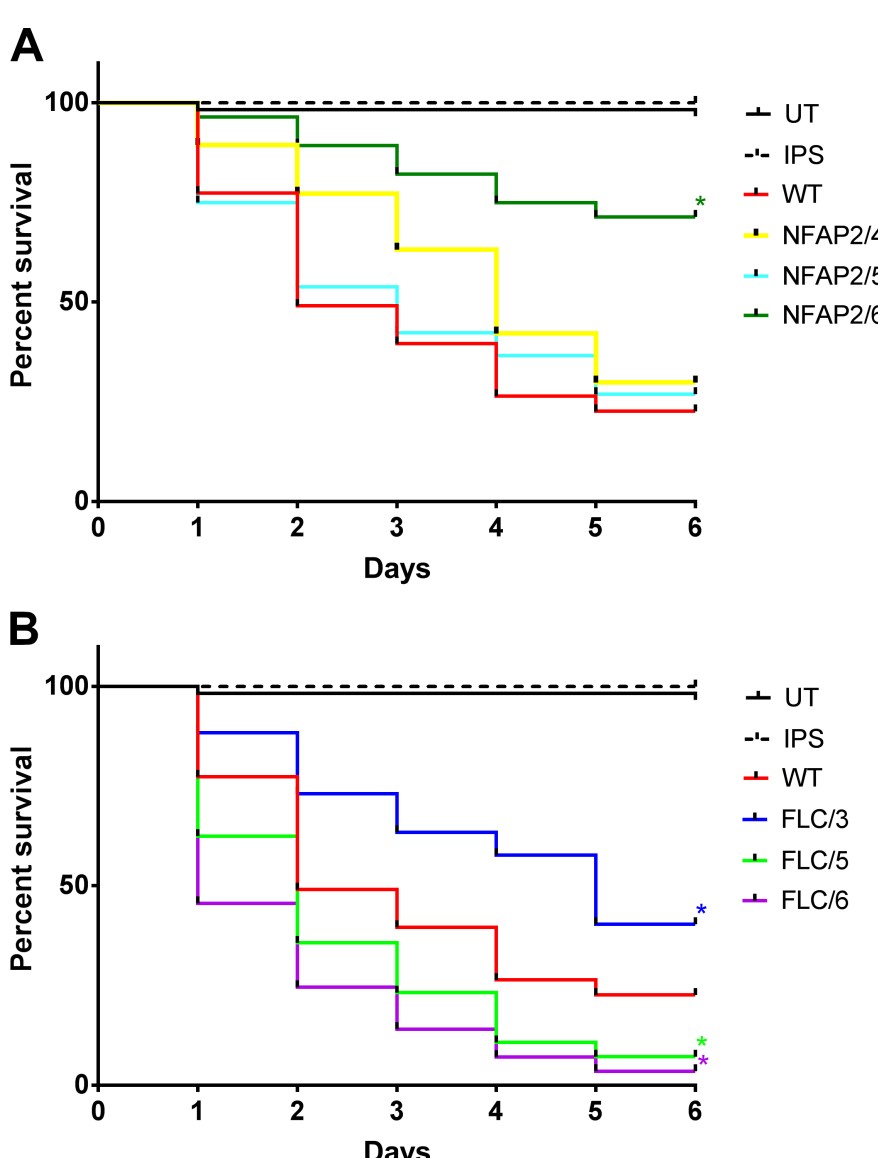

**FIG 5** Survival of *G. mellonella* larvae after infection with *C. albicans* strains tolerant to NFAP2 (NFAP2/4, 5, 6) (A) and resistant to FLC (FLC/3, 5, 6) (B) compared with the parental wild-type CBS 5982 (WT) strain. UT: untreated control and IPS: insect physiological saline-treated control. *$P \leq 0.05$ from both log-rank (Mantel–Cox) and Gehan–Breslow–Wilcoxon tests.

Studies have reported that genes involved in FLC resistance are typically related to ergosterol biosynthesis (e.g., ERG11 and ERG3), efflux pumps (e.g., CDR1, ABC1, and MDR1), chaperones (e.g., HSP90), and transcription factors (e.g., UPC2 and TAC1) in *C. albicans* (37, 48). Our data obtained with the evolved FLC-resistant strains are not consistent with these reports since we did not observe mutations in any of the genes mentioned above. The reason of this inconsistency could be the medium and fungal strain differing from those used in published microevolution experiments to generate FLC resistance in *C. albicans*. However, the mutated genes in our study (CHC1, PTC2, INP51, and RIM101) could provide resistance to FLC and could affect susceptibility to TRB, stress tolerance, metabolic fitness, and virulence in *C. albicans*, either directly or indirectly. The respective strain-related phenotypes are discussed below.

FLC/3 contains two SNVs, a nucleotide insertion in CHC1, and two nucleotide deletions in PTC2 (Table 1). Chc1p is a structural subunit of eukaryotic clathrin (49). Clathrin-coated vesicles play a focal role in transfer between cellular compartments;

furthermore, clathrin-mediated endocytosis (CME) is the primary mechanism of endocytosis in eukaryotic cells (50). In FLC/3, the nucleotide insertion in CHC1 creates a premature termination codon, which results in a partially translated protein without conserved clathrin heavy chain repeat (CLH; smart00299) and clathrin (pfam00637) domains (Fig. S2). Therefore, the translated protein is most likely dysfunctional, and CME is impaired. Rollenhagen et al. produced a strain of clathrin-deficient *C. albicans* that was resistant to FLC; therefore, we hypothesize that a heterozygous mutation in CHC1 can lead to FLC resistance in FLC/3 because the cellular internalization of FLC is required for its antifungal effect (51). Several publications have indicated that in yeast, clathrin deficiency causes abnormal morphology, slow growth, reduced endocytosis, and weakened cell function, but is not lethal (52–55). Our observations regarding FLC/3 can support CHC1 dysfunction because this strain has a highly wrinkled cell surface (Fig. 2A), reduced fitness (Fig. 4), heat tolerance (Fig. 3C), and virulence (Fig. 5B). Ptc2p plays a specific role in $CO_2$-responsive hyphal elongation in *C. albicans* (56). In FLC/3 both nucleotide deletions in PTC2 create a premature termination codon resulting in the loss of 40 C-terminal amino acids of protein (Fig. S2). The absence of the native C-terminus can potentially compromise the functionality of the protein. $CO_2$ is a crucial factor for target location and filamentation in *C. albicans* during host infection when the physiological concentration of $CO_2$ can reach several times higher values than that of atmospheric $CO_2$ (57, 58). Deletion of PTC2 in *C. albicans* caused a defect in $CO_2$-induced hyphal elongation at physiological concentrations of $CO_2$ (56). Mutant Ptc2p, presumed to have diminished function in FLC/3, may contribute to reduced virulence together with the less effective CME (Fig. 5B).

FLC/5 shows one SNV in a conserved domain region of INP51, which is similar to the catalytic inositol polyphosphate 5-phosphatase (INPP5c) domain of *S. cerevisiae* (INPP5c_ScInp51p-like; cd09090; Fig. S2). This SNV results in the exchange of a small nonpolar glycine into a hydrophobic valine in the translated protein (Fig. S2), which can considerably change the physicochemical properties of the protein. Inp51p regulates phosphatidylinositol 4,5-bisphosphate level, maintains cell wall integrity under stress and contact-induced hyphal growth, and may therefore regulate virulence (59, 60). Disruption of Inp51p function in *C. albicans* caused cell wall defects, abnormal distribution of chitin, decreased contact-induced filamentation, and reduced virulence (60). Our observations with FLC/5 contradict these findings because this strain had a normal cell surface (Fig. 2A) and increased virulence in the *G. mellonella* infection model (Fig. 5B). These results indicate that the glycine-valine change (which increases the overall hydrophobicity) in INP51 does not disrupt the function of Inp51p but might result in gain-of-function in FLC/5. Considering that FLC causes cell wall stress indirectly through membrane distribution, this gain-of-function mutation may explain the increased FLC tolerance of this strain because it can compensate for the FLC-induced cell wall defect better than WT (61).

In FLC/6, one SNV is present in a non-conserved domain region of RIM101. This mutation leads to a hydrophobic tyrosine to a hydrophobic cysteine amino acid exchange (Fig. S2). Rim101p, as a member of the Rim pathway, regulates antifungal drug tolerance proteins, such as Hsp90p and Ipt1p (62–65). Deletion of RIM101 increased susceptibility to azoles and echinocandins in *C. albicans* (66). We speculate that the mutation increases the activity of Rim101p, which could explain the FLC resistance of FLC/6.

Studies on the genetic background of tolerance and/or resistance to AFPs are lacking. All studies agree that due to the multimodal mode of action of AFPs (38, 67), resistance mechanisms evolve more slowly and require multiple gene mutations compared with conventional antifungal drugs (68, 69). For example, the simultaneous accumulation of mutations in different genes was found to drive NaD1 tolerance in *S. cerevisiae* because single mutations cannot lead to the observed NaD1-tolerant phenotype (38).

The findings of our study show that NFAP2/4 comprises a deletion of three nucleotides in a UBIP (GenBank: KHC66060.1) with unknown function and no conserved

domains (Fig. S2). This deletion causes the loss of a polar serine. Considering that this strain showed decreased tolerance to abiotic stress compared with the WT, this gene is assumed to be directly or indirectly involved in stress responses. The genomes of NFAP2/5 and NFAP2/6 contain identical MNVs at the same position in PGA58 (Table 1) with no conserved domain (Fig. S2). These MNVs result in an amino acid change from a hydrophobic valine to a small nonpolar alanine. Pga58p is involved in cell wall biogenesis, virulence, and biofilm formation in *C. albicans* (70–72). Because none of the mutations in UBIP and PGA58 in NFAP2-tolerant strains are nonsense mutations, the dysfunction of these proteins can be ruled out. All NFAP2-tolerant strains showed reduced NFAP2 uptake (Fig. S4; Table 3), no change in metabolic fitness (Fig. 4), and no increase in virulence (Fig. 5A). Reduced NFAP2 uptake could be caused by changes in the composition and function of the cell wall. Although the NFAP2-tolerant strains showed the same susceptibility to NFAP2 in the microdilution assay, the increased cell death of NFAP2/4 after NFAP2 treatment according to FACS analysis (Table 3) and strongly reduced virulence of NFAP2/6 compared with other NFAP2-tolerant strains (Fig. 5A) suggested that *C. albicans* has different ways to develop NFAP2 tolerance, which could not be exclusively correlated to the gene mutations observed. These nongenetic processes can be responses to AFP pressure to help overcome its harmful effects, as reported by Fitzgerald-Hughes et al., who described numerous proteomic changes in histatin 3-resistant derivatives of *C. albicans* compared with the parental strain (40). This hypothesis requires further investigation (e.g., transcriptome and proteome analyses) to understand the processes involved in NFAP2 tolerance.

According to our previous studies, NFAP2 disrupts the plasma membrane at MIC, but it does not induce apoptosis in yeast (73). Pavela et al. reported that NFAP2 selectively binds to phosphatidylinositol 5-phosphate and phosphatidic acid (74). Thus, NFAP2 may have cell membrane targets. The slightly increased susceptibility of the NFAP2-tolterant cells to FLC (Table 2) can indicate the possibility that the membrane composition of NFAP2-tolerant strains has changed in a way that is beneficial against NFAP2 binding but makes the cells more susceptible to FLC since FLC also causes membrane stress (21). Interestingly, the FLC-resistant strains are less susceptible to NFAP2 compared to the WT (Table 2), which also points to a possible involvement of a membrane target of NFAP2. However, in this study, we observed NFAP2 internalization in *C. albicans* cells treated with sublethal concentrations of NFAP2 (Fig. S4), indicating that NFAP2 may have intracellular targets. Studying the exact antifungal mechanism of NFAP2 and identification of its molecular target(s) in *C. albicans* can help explain the processes involved in NFAP2 tolerance and the observed phenotypes.

## Conclusion

In this study, we developed tolerance to 1× MIC of NFAP2 and resistance to 32× MIC of FLC in *C. albicans in vitro*. Genome analysis revealed non-silent mutations in only two genes (UBIP and PGA58) that might be involved in NFAP2 tolerance. FLC-resistant strains carry mutations in several genes (CHC1, INP51, PTC2, and RIM101), which probably contribute directly or indirectly to resistance and the observed phenotypes. The development of resistance to an antifungal agent leads to changes in susceptibility to other antifungal compounds, fitness, stress tolerance, and virulence. The development of tolerance to NFAP2 increased susceptibility to FLC and decreased susceptibility to other AFPs in *C. albicans*. However, the metabolic fitness of NFAP2-tolerant *C. albicans* remained unchanged, and no increase in virulence was detected compared with FLC-resistant strains. Therefore, *C. albicans* has limited ability to develop strong resistance mechanisms to NFAP2 compared with FLC. NFAP2-based topical anti-*Candida* therapy should be considered, as rapid development of strong resistance may not occur in a short period of time.

## MATERIALS AND METHODS

### Antifungal protein production

Recombinant NFAP2, PAF, PAFγ$^{opt}$, PAFB, and PAFC were produced in a *P. chrysogenum*-based expression system (75) and purified, as described (17, 27–29).

### Microevolution experiment to develop resistance to NFAP2 and FLC

To generate NFAP2- and FLC-resistant strains of *C. albicans* CBS 5982, the method described by Papp et al. was applied with several modifications (26). The microevolution experiment was performed in LCM (glucose 5 g L$^{-1}$, yeast extract 0.25 g L$^{-1}$, and peptone 0.125 g L$^{-1}$), supplemented with NFAP2 or FLC (MedChemExpress; Monmouth Junction, NJ, USA). Six independent microevolution experiments were performed in parallel for both. In the adaptation period, mid-log phase *C. albicans* cells ($2 \times 10^5$ cells/mL) cultured in LCM overnight at 30°C with continuous shaking (160 rpm) were inoculated in a 1:500 vol ratio of LCM supplemented with 0.5× MIC NFAP2 or FLC and incubated for 24 h (30°C, 160 rpm). The culture was inoculated in a 1:5,000 vol ratio in the same medium and incubated for 24 h two more times. The *Candida* cell cultures were then inoculated in 1:500 vol ratio, and the same subculturing steps were applied under the same conditions, but the concentration of NFAP2 or FLC was doubled in the medium (1× MIC). Serial passage was repeated until the highest antifungal concentration (maximum 32× MIC) was reached where *Candida* cells could grow. This resulted in *Candida* cell lines that adapted to NFAP2 or FLC. The growth of *Candida* cells was monitored by measuring the optical density (OD$_{600}$) of the cultures at the end of the third subculturing step of each antifungal concentration (Genesys 150 UV-Visible Spectrophotometer; Thermo Fischer Scientific, Waltham, MA, USA). In the next evolution period, cultures with the highest concentration of NFAP2 or FLC that *Candida* adapted to were inoculated in a 1:500 vol ratio of LCM without NFAP2 or FLC and incubated for 24 h (30°C, 160 rpm). Then cultures were inoculated in a 1:500 vol ratio of LCM without NFAP2 or FLC and incubated for 24 h. This step was repeated eight more times. Subsequently, 10-fold serial dilutions ($10^{-1}$–$10^{-6}$) of cultures from the last steps were prepared, and 5 µL from the $10^{-4}$–$10^{-6}$ dilution steps were streaked on 5 mL LCM agar plates (1.5% m/V; TC Plate 6 Well, Suspension, F; Sarstedt, Nümbrecht, Germany) supplemented with the highest concentrations of NFAP2 or FLC at which *C. albicans* CBS 5982 could grow. The plates were incubated for 48 h at 30°C, then one colony each from the six cell lines was isolated and maintained under selective conditions on LCM agar plates. This phase resulted in six independently evolved NFAP2-tolerant and FLC-resistant *C. albicans* cell lines. The schematic representation of the microevolution experiment is presented in supplementary material (Fig. S5). As control, *C. albicans* CBS 5982 was subjected to the same experimental setup, but in the absence of NFAP2 and FLC.

### Whole-genome sequencing and data analysis

Genomic DNA was isolated from three mutants, non-selected control, and parental WT strains using Blood and Cell Culture DNA Maxi Kit (QIAGEN, Hilden, Germany) according to the manufacturer's instructions.

Indexed sequencing libraries were generated from 1 ng genomic DNA using Nextera XT DNA Library Preparation Kit (Illumina, San Diego, CA, USA) following the protocol of the manufacturer. Libraries were validated and quantitated by capillary gel electrophoresis with a 2100 Bioanalyzer instrument (Agilent, Santa Clara, CA, USA), pooled and denatured with 0.1 M NaOH, loaded in MiSeq Reagent Kit v2-300 (Illumina, San Diego, CA, USA) at a concentration of 10 pM, and sequenced in a MiSeq DNA sequencing instrument (Illumina, San Diego, CA, USA), generating $2 \times 150$ bp sequence reads. Primary sequence analysis (base calling, demultiplexing, and fastQ file generation) was performed using Illumina RTA 1.18.54 and GenerateFASTQ 1.1.0.64. The genome of WT *C. albicans* CBS 5982 was assembled with SPAdes 3.13.0 (76) with the help of a reference

*C. albicans* SC5314 haploid genome (GCA_000182965.3). Genes were predicted with GlimmerHMM (23) and Funannotate (https://github.com/nextgenusfs/funannotate) tools (25).

Variant analysis was performed with CLC Genomics Workbench version 21.0.3 (https://digitalinsights.qiagen.com/). After quality control of reads was complete, the Basic Variant Detection tool was used with default settings. The assembled WT *C. albicans* CBS 5982 genome was used as a reference. Variants with <30% prevalence were filtered out. The variants in WT were removed, and silent mutations were not considered for analysis. Identified nonsynonymous mutations in exogenic regions based on the GlimmerHMM gene predictions are listed in Supplementary File S1, and nonsynonymous mutations in exogenic regions based on the Funannotate gene predictions are listed in Supplementary File S2.

The acquired gene sequences carrying mutations (Supplementary Files S1 and S2) were identified using the BLAST tool available on the National Center for Biotechnology Information website (https://blast.ncbi.nlm.nih.gov/Blast.cgi). The resulting gene matches from the *C. albicans* SC5314 genome database with the highest percent identity score above the 90% threshold value were considered identical to the queries (Table 1). Because the reference *C. albicans* SC5314 genome (GCA_000182965.3) used in the assembly of the WT *C. albicans* CBS 5982 genome is available as haploid, in certain cases, variant analysis may recognize native heterozygous loci of the diploid WT genome as mutations. To confirm the presence and heterozygosity of the mutations resulting from genome analyses, the affected gene regions were amplified from the genomic DNA of WT and the tolerant and resistant strains by PCR (Table S1). The amplicons were subjected to Sanger sequencing (Eurofins Genomics; Ebersberg, Germany), and the obtained sequences were compared with WT sequences through sequence alignment in CLC Genomics Workbench version 21.0.3 (https://digitalinsights.qiagen.com/) software (Sci Ed Software LLC, Westminster, CO, USA). The genes indicated as mutants by WGS were omitted when the amplicon analyses of the WT and the claimed mutant gene regions denied the presence of mutations (highlighted in red in Supplementary Files S1 and S2). Gene mutations that were unique to the tolerant or resistant strains and absent in non-selected control lines were considered a consequence of the NFAP2 or FLC pressure (omitted mutations from non-selected control lines are highlighted in yellow in Supplementary Files S1 and S2). The conserved domain regions of these mutant genes were identified and subjected to functional analysis by using Conserved Domain Search (https://www.ncbi.nlm.nih.gov/Structure/cdd/wrpsb.cgi; applying default options with "Standard display" result mode) and Conserved Domain Database (https://www.ncbi.nlm.nih.gov/cdd/) (77). Sequence files provided by Eurofins Genomics can be found in Supplementary File S3.

## Cell morphology investigation by scanning electron microscopy

Mid-log phase *C. albicans* cells ($4 \times 10^6$) were treated with 3.125 µg mL$^{-1}$ NFAP2 in LCM for 30 min at 30°C under continuous shaking at 160 rpm. Untreated cells served as morphology control. Cells were harvested ($9,000 \times g$ for 5 min), washed two times, and resuspended in phosphate-buffered saline (PBS). Then 8 µL samples were spotted on a silicon disk coated with 0.01% (wt/vol) poly-L-lysine (Merck Millipore, Billerica, MA, USA). Cells were fixed with 2.5% (vol/vol) glutaraldehyde and 0.05 M cacodylate buffer (pH 7.2) in PBS overnight at 4°C. The disks were washed two times with PBS and dehydrated with a graded ethanol series (30%, 50%, 70%, 80%, and 100% ethanol [vol/vol] for 24 h each at 4°C). The samples were dried with a Quorum K850 critical-point dryer (Quorum Technologies, Laughton, UK), coated with 12 nm gold, and observed with a JEOL JSM-7100F/LV field emission scanning electron microscope (JEOL Ltd., Tokyo, Japan).

## Antifungal susceptibility testing

According to Tóth et al. (19), broth microdilution susceptibility tests were performed with conventional antifungal drugs AMB (Santa Cruz Biotechnology, Dallas, TX, USA), FLC (Sigma-Aldrich, St. Louis, MO, USA), MFG (MedChemExpress, Monmouth Junction, NJ, USA), and TRB (MedChemExpress, Monmouth Junction, NJ, USA), as well as AFPs, such as NFAP2, PAF, PAFγ$^{opt}$, PAFB, and PAFC, to determine their MICs in LCM against 2 × 10$^4$ cells of WT *C. albicans* CBS5982 and strains tolerant to NFAP2 and resistant to FLC. Final concentrations were 16–0.0156 μg mL$^{-1}$ for AMB and MFG; 256–0.0156 μg mL$^{-1}$ for TRB; 8,192–0.0625 μg mL$^{-1}$ for FLC; and 50–1.562 μg mL$^{-1}$ for NFAP2, PAF, PAFγ$^{opt}$, PAFB, and PAFC in twofold dilutions. The cultures were resuspended by pipetting in microtiter plates (TC Plate 96 Well, Suspension, F; Sarstedt, Nümbrecht, Germany) and incubated at 30°C for 48 h without shaking. OD$_{600}$ values were measured with a microplate reader (Thermo Labsystems 354 Multiskan Ascent Microplate Reader; Thermo Fischer Scientific, Waltham, MA, USA) after suspending the cells in each well. MIC was defined as the lowest concentration of an antifungal compound that reduces fungal growth to 10% compared with the untreated control, which was set to 100%. Antifungal susceptibility tests were repeated at least two times, including two technical replicates.

## NFAP2 localization studies

For NFAP2 localization in *C. albicans*, NFAP2 was labeled with the green fluorophore BODIPY FL-EDA (Bd; Invitrogen, Waltham, MA, USA) according to Sonderegger et al. (78). Mid-log phase *C. albicans* cells (4 × 10$^6$) were treated with 2.5 μg mL$^{-1}$ Bd-NFAPP2 in LCM for 30, 60, 120, and 180 min at 30°C with continuous shaking at 160 rpm, washed with PBS, and incubated with 5 μg mL$^{-1}$ PI (Sigma-Aldrich, St. Louis, MO, USA) for 10 min at room temperature in the dark, washed two times with PBS, and resuspended in PBS. The samples were examined with a confocal laser (488 nm) scanning microscope (Olympus Fluoview FV1000; Olympus, Shinjuku, Japan). Excitation and emission wavelengths were 504 and 512 nm for NFAP2-Bd and 535 and 617 nm for PI, respectively. Sequential scanning was used to avoid crosstalk between the fluorescent dyes.

## NAFP2-uptake analysis

NFAP2 uptake by *C. albicans* strains was analyzed by FACS. *C. albicans* cells (4 × 10$^6$) were treated with 3.125 μg mL$^{-1}$ Bd-NFAP2 in LCM (30°C, 16 h, 160 rpm) and costained with PI (as described for NFAP2 localization). Bd-NFAP2- and PI-positive cells were detected by a FlowSight imaging flow cytometer equipped with lasers at 405 (violet), 488 (blue), and 642 nm (red; Amins, Merck Millipore, Billerica, MA, USA). Calibration controls were used to avoid overexposure of the positive events in specific fluorescent channels, which cause false positive staining signals in other fluorescent channels. Cells treated with 70% (vol/vol) ethanol (10 min at room temperature, 160 rpm) were used as positive PI staining and calibration controls, and cells treated with 2 μg mL$^{-1}$ fluorescein diacetate (Thermo Fischer Scientific, Waltham, MA, USA; 20 min at room temperature, 160 rpm) were used as Bd-NFAP2 calibration controls. Ten thousand cells per run were detected. Bd-NFAP2 was detected at 488 nm and PI at 642 nm, with excitation lasers and emission in channel 2 window. Gating was adjusted to reach at least 96% of the untreated cells, and debris was excluded during data acquisition. Data analysis was performed with Image Data Exploration and Analysis software (IDEAS; Amins, Millipore, Billerica, MA, USA). The FACS experiments were repeated three times.

## Stress tolerance analyses

For stress tolerance assays, *C. albicans* strains were grown overnight in LCM (30°C, 160 rpm) to reach the mid-log phase, the OD$_{600}$ of the cultures was set at 0.1, and fivefold dilution series were prepared in the same medium. Five microliter from the dilution series was spotted on the surface of LCM agar plates and dried before incubation at 30°C for 24 h. After the incubation period, the plates were photographed (Versa Doc

Imaging System 4000 MP; Bio-Rad, Hercules, CA, USA). To investigate tolerance to NaCl, SDS, and CFW, agar plates were supplemented with serial dilutions of CFW (1, 2.5, 5, and 10 µg mL$^{-1}$), SDS (12.5, 25, 50, and 100 µg mL$^{-1}$), or NaCl (50, 100, 200, and 300 mM). To investigate tolerance to UV irradiation, the plates were irradiated with UV light for 0, 0.5, 1, 2, and 5 min with the germicidal lamp of a laminar flow box (Herasafe; Thermo Electron Corporation, Waltham, MA, USA). To investigate tolerance to heat stress, the fivefold dilution series of *C. albicans* strains were heat-treated at 30°C, 37°C, 40°C, and 45°C for 30 min. Two technical replicates were prepared for all stress tolerance analyses, and tests were repeated at least two times.

## Metabolic fitness assays

Metabolic adaptation and growth of WT CBS 5982 and NFAP2-tolerant and FLC-resistant strains of *C. albicans* were monitored in LCM, Vogel's medium (10 g L$^{-1}$ D-glucose, 2.5 g L$^{-1}$ Na$_3$-citrate, 5 g L$^{-1}$ anhydrous KH$_2$PO$_4$, 2 g L$^{-1}$ anhydrous NH$_4$NO$_3$, 0.2 g L$^{-1}$ MgSO$_4$·7H$_2$O, 0.1 g L$^{-1}$ CaCl$_2$·2H$_2$O, 5.26 mg L$^{-1}$ monohydrous citric acid, 5.26 mg L$^{-1}$ ZnSO$_4$·7H$_2$O, 1.05 mg L$^{-1}$ Fe[NH$_4$]$_2$[SO$_4$]·6H$_2$O, 0.26 mg L$^{-1}$ CuSO$_4$·5H$_2$O, 0.05 mg L$^{-1}$ MnSO$_4$·4H$_2$O, 0.05 mg L$^{-1}$ H$_3$BO$_3$, 0.05 mg L$^{-1}$ Na$_2$MoO$_4$·2H$_2$O, and 0.05 mg L$^{-1}$ biotin), ME (Sigma-Aldrich, St. Louis, MO, USA), and RPMI-1640 (Gibco - Thermo Fischer Scientific, Waltham, MA, USA) media and compared with that of WT. Next, 200 µL of mid-log phase *C. albicans* cells (OD$_{600}$ = 0.1) was inoculated in 20 mL of the respective media and incubated for 18 h at 30°C and 160 rpm. OD$_{600}$ values of the cultures were measured every hour with a Genesys 150 UV-Visible Spectrophotometer (Thermo Fischer Scientific, Waltham, MA, USA). The metabolic fitness assays were repeated three times.

## Virulence assay

A *G. mellonella in vivo* infection model was used to investigate the virulence of the generated NFAP2-tolerant and FLC-resistant strains compared with the parental WT CBS 5982 strain. Twenty *G. mellonella* larvae (TruLarv; BioSystems Technology, Exeter, United Kingdom) were infected with 20 µL of $5 \times 10^7$ *C. albicans* cells per milliliter suspended in insect physiological saline (IPS: 50 mM NaCl, 5 mM KCl, 10 mM EDTA, and 30 mM sodium citrate in 0.1 M Tris-HCl [pH 6.9]) by intrahemocoelic injection (29-gauge insulin needles; BD Micro-Fine, Franklin Lakes, NJ, USA) through the last proleg. IPS-treated larvae served as the uninfected controls, whereas larvae without interventions served as untreated controls. The larvae were incubated at 37°C, and survival was monitored every 24 h for 6 d. The virulence assay was repeated three times.

## Statistical analyses

For FACS analysis, a generalized linear regression model with binomial distribution was applied to the data due to the categorical nature of the response variable (glm() function). Models were evaluated using type II ANOVA. Pairwise comparison with Bonferroni's correction was used for post hoc analysis in cases of significant effects. During pairwise comparison, estimated marginal means were tested against the data set of the WT; $P$ values $\leq 0.05$ were considered significant ($n = 30,000$ per strain). To assess the fitness curves of the different strains, repeated measure ANOVA was used (fitrm() and ranova() functions), and a different repeated measure model was fitted for each substrate. In the case of significant models, the Tukey–Kramer post hoc test was used for comparison; $P$ values $\leq 0.05$ were considered significant. MATLAB 2022b (MathWorks, Portola Valley, CA, USA) and R (R Core Team [2022] R: A Language and Environment for Statistical Computing, R Foundation for Statistical Computing, Vienna) software were used to evaluate FACS and fitness test data. To compare survival curves in *G. mellonella* larval infection model experiments, log-rank (Mantel–Cox) and Gehan–Breslow–Wilcoxon tests were used in GraphPad Prism 7.00 (GraphPad Software, Boston, MA, USA). Survival was considered significant if $P \leq 0.05$ in both tests, $n = 60$ per strain.

## ACKNOWLEDGMENTS

The present work of L.G. was financed by the Hungarian National Research, Development and Innovation Office—NKFIH, FK 134343, and K 146131 projects. The research was funded in part by the Austrian Science Fund FWF (I3132-B21) to F.M. University of Szeged Open Access Fund, Grant ID: 7051. This article/publication is based upon work from COST Action EURESTOP, CA21145, supported by COST (European Cooperation in Science and Technology).

## AUTHOR AFFILIATIONS

[1]Department of Biotechnology, Faculty of Science and Informatics, University of Szeged, Szeged, Hungary

[2]Doctoral School of Biology, Faculty of Science and Informatics, University of Szeged, Szeged, Hungary

[3]Department of Biochemistry and Molecular Biology, Faculty of Science and Informatics, University of Szeged, Szeged, Hungary

[4]Institute of Plant Biology, HUN-REN Biological Research Center, Szeged, Hungary

[5]Department of Microbiology, Faculty of Science and Informatics, University of Szeged, Szeged, Hungary

[6]Department of Physiology, Albert Szent-Györgyi Medical School, University of Szeged, Szeged, Hungary

[7]Institute of Biophysics, HUN-REN Biological Research Center, Szeged, Hungary

[8]Institute of Molecular Biology, Biocenter, Medical University of Innsbruck, Innsbruck, Austria

[9]Institute of Biochemistry, HUN-REN Biological Research Center, Szeged, Hungary

## AUTHOR ORCIDs

Gábor Bende http://orcid.org/0000-0003-4571-9775
Gábor Rákhely http://orcid.org/0000-0001-5616-7385
Florentine Marx http://orcid.org/0000-0002-8408-1842
László Galgóczy http://orcid.org/0000-0002-6976-8910

## FUNDING

| Funder | Grant(s) | Author(s) |
| --- | --- | --- |
| Nemzeti Kutatási Fejlesztési és Innovációs Hivatal (NKFI) | FK 134343,K 146131 | László Galgóczy |
| Austrian Science Fund (FWF) | I3132-B21 | Florentine Marx |
| University of Szeged Open Access Fund | ID: 7051 | László Galgóczy |

## AUTHOR CONTRIBUTIONS

Gábor Bende, Formal analysis, Investigation, Validation, Visualization, Writing – original draft | Nóra Zsindely, Investigation, Methodology, Validation | Krisztián Laczi, Data curation, Methodology, Validation, Visualization | Zsolt Kristóffy, Investigation | Csaba Papp, Investigation, Methodology | Attila Farkas, Investigation, Visualization | Liliána Tóth, Investigation | Szabolcs Sáringer, Formal analysis, Methodology, Validation | László Bodai, Conceptualization, Investigation, Methodology, Writing – original draft | Gábor Rákhely, Resources, Writing – original draft | Florentine Marx, Conceptualization, Funding acquisition, Resources, Writing – original draft | László Galgóczy, Conceptualization, Funding acquisition, Methodology, Project administration, Resources, Supervision, Visualization, Writing – original draft, Writing – review and editing

## DATA AVAILABILITY

The raw reads were uploaded to the European Nucleotide Archive under the bio project No.: PRJEB66720.

## ADDITIONAL FILES

The following material is available online.

### Supplemental Material

**Supplemental file S1 (Spectrum01273-24-s0001.xlsx).** Identified nonsynonymous mutations in exogenic regions in generated *C. albicans* strains.
**Supplemental file S2 (Spectrum01273-24-s0002.xlsx).** Identified nonsynonymous mutations in exogenic regions in generated *C. albicans* strains.
**Supplemental file S3 (Spectrum01273-24-s0003.txt).** Sequences of the mutated gene regions in comparison with that of the wild-type.
**Supplemental figures and tables (Spectrum01273-24-s0004.docx).** Table S1 and S2; Figure S1 to S5.

### Open Peer Review

**PEER REVIEW HISTORY (review-history.pdf).** An accounting of the reviewer comments and feedback.

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
