## [Reviewer comments · Microbiology Spectrum]

Microbiology Spectrum

The *Neosartorya (Aspergillus) fischeri* antifungal protein NFAP2 has low potential to trigger resistance development in *Candida albicans in vitro*

Gábor Bende, Nóra Zsindely, Krisztián Laczi, Zsolt Kristóffy, Csaba Papp, Attila Farkas, Liliána Tóth, Szabolcs Sáringér, László Bodai, Gábor Rákhely, Florentine Marx, and László Galgóczy

Corresponding Author(s): László Galgóczy, University of Szeged, Faculty of Science and Informatics

Review Timeline:

Submission Date:	May 23, 2024
Editorial Decision:	July 14, 2024
Revision Received:	September 30, 2024
Accepted:	October 28, 2024

Editor: Gustavo Goldman

Reviewer(s): Disclosure of reviewer identity is with reference to reviewer comments included in decision letter(s). The following individuals involved in review of your submission have agreed to reveal their identity: Norman van Rhijn (Reviewer #1)

Transaction Report:

DOI: <https://doi.org/10.1128/spectrum.01273-24>

Re: Spectrum01273-24 (The *Neosartorya (Aspergillus) fischeri* antifungal protein NFAP2 has low potential to trigger resistance development in *Candida albicans in vitro*)

Dear Dr. László Galgóczy:

Thank you for the privilege of reviewing your work. Below you will find my comments, instructions from the Spectrum editorial office, and the reviewer comments.

Your manuscript has been reviewed by two reviewers. Please, address their suggestions and comments by submitting a revised version together with a rebuttal letter.

Revision Guidelines

Sincerely,
Gustavo Goldman
Editor
Microbiology Spectrum

Reviewer #1 (Comments for the Author):

The article "The *Neosartorya (Aspergillus) fischeri* antifungal protein NFAP2 has low potential to trigger resistance development in *Candida albicans in vitro*" describes the anti fungal protein NFAP2 from *A. fischeri* by looking at resistance development in *Candida albicans*. Antifungal proteins are of high interest and resistance to them would be important to understand for the

further development of a novel anti fungal.

I have several major comments which have to do with the results and experimental design.

- It is widely understood that resistance means the shift of MICs. Still whole article present "resistant strains" However, the MIC does not shift significantly (a shift of 4 fold is generally considered significant, while 2 can be due to other factors).
- I believe it is more likely that this is persistence or tolerance, and think it would be better to write the whole article exploring this phenomenon as it is clearly not resistance.
- The FACS analysis is nowhere to be found.
- In several experiment the wording "significant" is used without any statistical analysis - the confocal microscopy and electron microscopy should be quantified.
- The difference between lineages shows that there is more than just the mentioned SNP. How many SNPs does each strain have in total. If a causation is implied, this should be reconstructed genetically in the parental isolate.
- The genome assembly of the CBS strain is presented in the methods but should also be in the results. How complete is it, how many genes, how does it compare?

Minor comments:

- L70-71 "due to antifungal drugs" patients does not increase because of this
- L80 "five belong to genus Candida" not any more since they have been renamed
- L81 "most common IFI" also not true, numbers wise it is cryptococcosis
- L104-105 "Among others, rezafungin (CD101), ibrexafungerp (SCY-078), and VT-1161 are the most promising candidates" This is subjective. If going by compounds that are furthest along in the pipeline Olorofim should be included too.
- L121 echinocandins has typo

Reviewer #2 (Comments for the Author):

In this study, authors demonstrated a potential ability of a previous identified antifungal protein (AFP-NFAP2) secreted by *Neosartorya* (*Aspergillus*) *fischeri* to trigger resistance against a fungal pathogen *C. albicans*. As a result, they found that NFAP2 has low potential to trigger resistance in *C. albicans* in vitro. Among three selected resistance isolates, the developed mechanisms by NFAP2 are not associated with severe phenotypic changes compared with development of resistance to generic FLC. In general, findings are very interesting, and data are able to support conclusions.

Comments:

- 1) In result 1 for "adaptation analysis in the microevolution experiment" if there is a design model, it should be easy to be understood.
- 2) Does inoculated pool amount is enough to screen? Only select three colonies of independently evolved *C. albicans* cultures resistant to NFAP2 and FLC are enough to represent, what is rational? Please discuss them.
- 3) Sequences of mutation-affected regions of these genes were checked for heterozygosity and compared with WT sequences. Please explain how to affect the susceptibility in the heterozygosity way.
- 4) Data indicated that development of resistance to NFAP2 does not influence or increase susceptibility to conventional antifungal drugs, except FLC. To this point, authors should discuss how and why?
- 5) In FIG. 1, there is no statistics analysis for data?
- 6) Section of discussion is too long, and some are not important.
- 7) In virulence test, they stated that NFAP2 resistance does not enhance the virulence of *C. albicans* compared with FLC resistance. However, in the curve of virulence, it does showing NFAP2 resistance affected virulence? Many curves mixed together.

University of Szeged
Faculty of Science and Informatics
Department of Biotechnology and Microbiology
Szeged, Közép fasor 52
H-6726, HUNGARY
www2.bio.u-szeged.hu/galgoczylab

Mail: Szeged, Közép fasor 52
H-6726, HUNGARY
Phone: +36 62 546 936
Fax: +36 62 544 352
galgoczi@bio.u-szeged.hu

Gustavo Henrique Goldman
Editor
Microbiology Spectrum

September 19, 2024

Dear Prof. Goldman,

We have revised our manuscript, Spectrum01273-24, entitled „*The Neosartorya (Aspergillus) fischeri antifungal protein NFAP2 has low potential to trigger resistance development in Candida albicans in vitro*” in accordance with the comments and suggestions provided by the reviewers. We are grateful for their thorough evaluation and constructive feedback, which enhanced the quality of our manuscript. We responded to all critics and hope that our manuscript now meets all criteria to be accepted for publication. We highlighted all changes in text by the track changes function of the word processor. This version is uploaded for review along with the clean copy of the manuscript (NFAP2 resistance manuscript_Bende G et al_R1_track changes.docx). Below, we provide a detailed, point-by-point response to each of the reviewers' suggestions, comments and criticisms.

Reviewer #1:

The article "The Neosartorya (Aspergillus) fischeri antifungal protein NFAP2 has low potential to trigger resistance development in Candida albicans in vitro" describes the anti fungal protein NFAP2 from A. fischeri by looking at resistance development in Candida albicans. Antifungal proteins are of high interest and resistance to them would be important to understand for the further development of a novel anti fungal. I have several major comments which have to do with the results and experimental design.

ANSWER: We thank the reviewer for the thorough review and for suggestions to improve the quality of our manuscript.

It is widely understood that resistance means the shift of MICs. Still whole article present "resistant strains" However, the MIC does not shift significantly (a shift of 4 fold is generally considered significant, while 2 can be due to other factors). I believe it is more likely that this is persistence or tolerance, and think it would be better to write the whole article exploring this phenomenon as it is clearly not resistance.

ANSWER: Thank you for this important criticism and recommendation. We agree that „tolerance” is the correct description of the phenomenon discussed. Therefore, we have corrected the manuscript accordingly.

The FACS analysis is nowhere to be found.

ANSWER: We apologize that we did not emphasize in the manuscript that the results of FACS analysis can be found in the „NFAP2 uptake analysis” subchapter (L249-262), and all related data are summarized in the Table 3. To make it clear for the readers, the title of Table 3 in the revised manuscript version emphasizes that all data is related to the FACS analysis.

In several experiment the wording "significant" is used without any statistical analysis¹ - the confocal microscopy² and electron microscopy³ should be quantified.

ANSWERS:

¹: We thank the reviewer for the remark, every word „significant” without statistical context has been rephrased in the revised manuscript version.

²: Our goal with confocal microscopy was to explore the timeline of NFAP2 binding to *C. albicans* cells. These results indicated detectable binding after 30 minutes under NFAP2 exposure and intracellular localization and cell death after 4 hours. We used FACS analysis to quantify the result of confocal microscopy, and to create statistics for NFAP2 binding and consequent cell death. For the FACS analysis we followed the protocol of the confocal microscopy experiment with the incubation time increased to 16 h.

³: Our electron microscopy results cannot be quantified, as the images were intended solely for visual evaluation of the cell morphology characteristic to evolved *C. albicans* strains.

The difference between lineages shows that there is more than just the mentioned SNP.¹ How many SNPs does each strain have in total.² If a causation is implied, this should be reconstructed genetically in the parental isolate.³ The genome assembly of the CBS strain is presented in the methods but should also be in the results.⁴ How complete is it, how many genes, how does it compare?⁴

ANSWERS:

¹: We agree with the reviewer’s comment, and we already addressed this in „Discussion” chapter (L429-436): „[...] *C. albicans* has different ways to develop NFAP2 tolerance, which could not be exclusively correlated to the gene mutations observed. [...] These nongenetic processes can be responses to AFP pressure to help overcome its harmful effects, [...] This hypothesis requires further investigation (e.g., transcriptome and proteome analyses) to understand the processes involved in NFAP2 tolerance.”

²: The process of identifying the mutations in the genomes of the NFAP2-tolerant and the FLC-resistant strains were the following: The Basic Variant Detection tool was used with default settings after the quality control of the reads. The assembled wild-type *C. albicans* CBS 5982 genome was used as a reference. Variants with <30% prevalence were filtered out. The variants in wild-type were removed, and silent mutations were not considered for analysis. Nonsynonymous mutations in exogenic regions were listed. This stage was the first point in the process, where we acquired a list of mutations. This list can be found in Supplementary File 1. At this stage the numbers of mutations including (SNPs) were NFAP2/4: 6 (SNPs: 4), NFAP2/5: 5 (SNP: 1), NFAP2/6: 5 (SNPs: 1), FLC/3: 6 (SNPs: 2), FLC/5: 3 (SNPs: 3), FLC/6: 2 (SNPs: 1), non-selected control line 1: 10 (SNPs: 4), non-selected control line 2: 10 (SNPs: 4), , non-selected control line 3: 13 (SNPs: 7). In the “Materials and methods” chapter we elaborated the further steps: “*Because the reference C. albicans SC5314 genome (GCA_000182965.3) used in the assembly of the WT C. albicans CBS 5982 genome is available as haploid, in certain cases, variant analysis may recognize native heterozygous loci of the diploid WT genome as mutations. To confirm the presence and heterozygosity of the mutations resulting from genome analyses, the affected gene regions were amplified from the genomic DNA of WT, and the tolerant and resistant strains by PCR (Table S1).*” The listed mutations in Supplementary File 1 were filtered via PCR verification by Sanger sequencing of the amplicons. The genes claimed to be carrying mutations were omitted when the PCR verification denied the presence of mutations (highlighted with red in Supplementary File 1). We presented the whole list in Supplementary File 1 for transparency. After the PCR verification process our final results were: NFAP2/4 in total has only one SNP in the UBIP (uncharacterized biofilm-induced protein) gene. NFAP2/5 and NFAP2/6 in total has the same multiple nucleotide variation (MNV) affecting two bases in the PGA58 gene. FLC/3 in total has two SNPs and an insertion in the CHC1 gene, and has two deletions in the PTC2 gene. FLC/5 in total has one SNP in the INP51 gene. FLC/6 in total has one SNP in the RIM101 gene. The detailed information can be found in Table 1, Table S1 and Supplementary File 1.

³: We agree with the reviewer’s comment; it would be advantageous to reconstruct the observed genomic changes in the parental strain to support our claims. However, genetic reconstruction by mutagenesis, especially heterozygous, is a substantial challenge and time-consuming in diploid eukaryotes like *C. albicans*. We believe, this experiment is out of our present scope as its outcome would not influence the results and conclusions of the manuscript. Our primary goal with genome analysis was to explore whether the NFAP2 tolerant strains possess unique mutations in the exogenic regions in comparison with the FLC-resistant and parental wild-type strains. The gene mutations present exclusively in the NFAP2-tolerant strains could be responsible for the tolerant phenotype according to data in the literature. Its possibility is discussed in the manuscript.

⁴: We are grateful for the reviewer for this important remark and recommendation, as based on this insightful comment, we realized that after genome sequencing and assembly of the CBS5982 wild-type strain, our gene prediction method using GlimmerHMM (Reference #23, Bioinformatics 2004;20:2878–2879) yielded 2983 genes compared to the ~6100 existing genes in a *Candida albicans* genome (see Genome Res. 2015;25(3):413-25.). Therefore, we performed an additional fungal-specific gene prediction using Funannotate (Palmer and Stajich, 2020 <https://github.com/nextgenusfs/funannotate>; doi: 10.5281/zenodo.1134477), which yielded 5951 genes.

The newly predicted genes were subjected to variant analysis as described in the „Materials and Methods/Whole genome sequencing and data analysis” chapter (L519-523). This variant analysis yielded novel gene mutations (genes with possible role in the developed resistances), and this analysis also re-confirmed those we had previously PCR-verified (see in the “Results/Whole genome sequencing and data analysis” chapter) and discussed in the first version of the manuscript based on the GlimmerHMM data. We added „Supplementary file 2” including the filtered list of the acquired gene mutations based on the gene predictions using Funannotate. The novel gene mutations were filtered as described in the first version of the manuscript in the “Materials and Methods/Whole genome sequencing and data analysis” chapter: mutations were disregarded if they were also present in the non-selected control strains. Furthermore, mutations were disregarded if their “Coverage” value was below 28 (30 with a -2 lenience). Finally, after the previously described filtering steps, we performed PCR verification of the mutations. The results confirmed that all of the predicted novel mutations in the NFAP2 tolerant and FLC resistant strains are native heterozygous loci, and they are also present in the wild-type strain, therefore, all novel mutations discovered by Funannotate were omitted.

The new analysis did not alter the results of the previously verified gene mutations from the first version of the manuscript. Therefore, we concluded that no modifications were necessary for the “Results” and “Discussion” sections. However, we added the new analysis tool “Funannotate” and its weblink and citation in the “Materials and Methods” (L517-518).

Additionally, we included the statistics of the genome assembly based on both gene prediction methods (GlimmerHMM and Funannotate) as requested by the reviewer, and added “Supplementary File 2”, which contains the list of gene mutations based on the variant analysis using the Funannotate gene prediction.

The requested information regarding the statistics of the genome sequencing and assembly:

The assembled CBS 5982 WT genome has 94.3% estimated genome completeness, 98.9% average nucleotide identity (ANI) with 72.4% coverage compared to the SC5314 reference genome.

The gene prediction using GlimmerHMM identified 2983 genes, and the gene prediction using Funannotate identified 5951 genes.

L70-71 "due to antifungal drugs" patients does not increase because of this.

ANSWER: We would like to respectfully point out, that the quoted sentence in L66-68 (L70-71 before revision) says “[...] *the occurrence of opportunistic fungal infections (FIs) has increased due the rise in the number of patients with immunosuppression and resistance to antifungal drugs.*”. We believe that this sentence is factually correct and is supported by the cited reference (Reference #1, J Fungi (Basel) 2022;8:946.).

L80 "five belong to genus *Candida*" not any more since they have been renamed.

ANSWER: We thank the reviewer for this valuable comment. We have corrected this sentence considering all *Candida* (and formerly *Candida*) species from all risk groups (from medium to high) indicated in the WHO fungal priority pathogens list. The corrected sentence is the following (L80-81): “*Four of the listed fungi belong to the genus Candida, while two was formerly classified in this genus (5).*”

L81 "most common IFI" also not true, numbers wise it is *Cryptococcus*.

ANSWER: We would like to kindly note that the sentence mentioned in L79-80 (L81 before revision) says “*Candidiasis [...] is the most common FI (6) [...]*”. FI abbreviates fungal infection in the manuscript. Here we are discussing the prevalence of the fungal infections (FIs) caused by *Candida* species generally, and not their prevalence in invasive fungal infections (IFIs). We believe that this sentence is correct in its present form, and it is supported by the cited reference (Reference #6, J Fungi 2017;3:57).

L104-105 "Among others, *rezafungin (CD101)*, *ibrexafungerp (SCY-078)*, and *VT-1161* are the most promising candidates" This is subjective. If going by compounds that are furthest along in the pipeline *Olorofim* should be included too.

ANSWER: We thank the reviewer for this input. The sentence discussed in L102-104 (L104-106 before revision) says “[...] *rezafungin (CD101)*, *ibrexafungerp (SCY-078)*, and *VT-1161* are the most promising candidates, which are highly active against several *Candida* species, including conventional drug-resistant *C. albicans* and *C. auris* isolates, [...]”. We would like to politely decline the suggestion to include *Olorofim*, as it is not effective against *Candida* species (see Reference #12, Open Forum Infect Dis 2020;7:ofaa016.), since our sentence was discussing new compounds with potent anti-*Candida* spectrum. On the other hand, we are grateful to the reviewer for bringing this paragraph to our attention: By mistake, we included *rezafungin*, which has recently been approved by the FDA (see J Infect Dis 2024;19:jiae146.), thus, it is no longer a clinical trial candidate. As further additions, instead of *rezafungin* we have included *fosmanogepix* in this paragraph, and for clarity, we have specified that the mentioned examples are phase III molecules. The revised version of this sentence (L102-104): “*Fosmanogepix (APX001)*, *ibrexafungerp (SCY-078)*, and

VT-1161 are among the most promising phase III clinical trial candidates, which are highly active against several Candida species, including conventional drug-resistant C. albicans and C. auris isolates (12)."

L121 echinocandins has typo.

ANSWER: We corrected "echinocandi/ns" to "echinocandins" in L119 (L121 before revision).

Reviewer #2:

In this study, authors demonstrated a potential ability of a previous identified antifungal protein (AFP-NFAP2) secreted by Neosartorya (Aspergillus) fischeri to trigger resistance against a fungal pathogen C. albicans. As a result, they found that NFAP2 has low potential to trigger resistance in C. albicans in vitro. Among three selected resistance isolates, the developed mechanisms by NFAP2 are not associated with severe phenotypic changes compared with development of resistance to generic FLC. In general, findings are very interesting, and data are able to support conclusions.

ANSWER: We thank the reviewer for the thorough review and for the suggestions to improve the quality of our manuscript.

In result 1 for "adaptation analysis in the microevolution experiment" if there is a design model, it should be easy to be understood.

ANSWER: We applied the method described previously by Papp et al. (2020) with several modifications. This method is cited as reference #26 in our manuscript (mSphere 2020;5:e00821-20). To clarify that we applied a modified protocol, we have added this reference to the related „Microevolution experiment to develop resistance to NFAP2 and FLC" subchapter of the "Materials and methods" chapter. Lastly, we would like to kindly bring FIG. S5 to the reviewer's attention, which is the schematic visual representation of the microevolution experiment for better understanding.

Does inoculated pool amount is enough to screen? Only select three colonies of independently evolved C. albicans cultures resistant to NFAP2 and FLC are enough to represent, what is rational? Please discuss them.

ANSWER: We would like to emphasize that creating large data pools was not our aim in this work. However, we agree with the reviewer's comment, that the overall analysis of three cell lines may not provide the highest statistical power and to create a general trend for the resistance development; but they can support the conclusions presented in the manuscript. Our main objective was to investigate whether the

development of resistance (tolerance) to NFAP2 could induce phenotypic and genetic changes compared to the FLC-resistant strains and the parental wild-type. Our findings confirmed these possibilities, as we identified various phenotypic changes in the NFAP2-resistant (tolerant) strains. Furthermore, we identified gene mutations unique to the NFAP2-resistant (tolerant) strains, as opposed to the FLC-resistant and non-selected control strains. We think that our experiments with three biological replicates in our study can provide valuable preliminary data that can guide future studies and help secure funding for more extensive research to reveal potential hotspot mutations related to resistance (tolerance)-development.

Sequences of mutation-affected regions of these genes were checked for heterozygosity and compared with WT sequences. Please explain how to affect the susceptibility in the heterozygosity way.

ANSWER: The impact of a heterozygous mutation on the phenotype depends on the nature of the gene and the mutation (see Reference #46, J Fungi 2019;6:10.). To determine how the mutation affects the susceptibility of the given strain to various antifungal agents, a series of complex and extensive analyses needs to be conducted, which is out of scope of present study. The type of the mutations needs to be determined, whether it is gain or loss of function. Usually, gain-of-function mutations typically result in a protein with enhanced or novel activity, while loss-of-function mutations often lead to a complete or partial loss of the protein's activity. To assess this, protein structure prediction methods, such as homology modeling or *ab initio* approaches, can be employed to analyze the impact of the amino acid changes on the protein's three-dimensional structure. Functional predictions can be made by molecular docking studies or molecular dynamics simulations to assess how the mutations might affect the protein's active sites, stability, or interaction with other molecules. Unfortunately, the aforementioned methods are highly resource- and time-intensive, and thus are beyond our current scope. We agree that it would be worth to investigate in the near future. Therefore, based on our findings, we are only able to conclude that the heterozygous mutations in the evolved strains might cause changes in antifungal drug susceptibility in *C. albicans*. We have already addressed this issue in "Discussion" (L348-350), however, we added a sentence which emphasizes the discussed phenomenon. Cited reference #47 supports this statement (Reference #47, Med Mycol 2018;56:687-694).

Data indicated that development of resistance to NFAP2 does not influence or increase susceptibility to conventional antifungal drugs, except FLC. To this point, authors should discuss how and why?

ANSWER: The exact mode of action and cellular targets of NFAP2 are unknown, thus, the consequences of *C. albicans* acquiring NFAP2 resistance (tolerance) are unpredictable at whole cell level. However, we know that NFAP2 disrupts the fungal cell membrane (Reference #73, AMB Express 2016;6:75, Reference #19, Front Microbiol 2018;9:393, Reference #17, Antimicrob Agents Chemother 2019;63:e01777-18),

therefore the background of the resistance (tolerance) can be changes in cell membrane composition. Since fluconazole (FLC) also causes membrane stress, it is possible that the membrane composition of NFAP2-resistant (tolerant) strains has changed in a way that is beneficial against NFAP2 binding but makes the cells more susceptible to FLC. This can be supported by our previous observations, that NFAP2 exerted synergy with FLC *in vitro* (Reference #19, Front Microbiol 2018;9:393, Reference #20, Int J Mol Sci 2021;22:771) and *in vivo* (Reference #17, Antimicrob Agents Chemother 2019;63:e01777-18) against *Candida* spp. Furthermore, our results show that, interestingly, the FLC-resistant strains are slightly less susceptible to NFAP2 compared to the wild-type. This suggests that the involvement of both molecules in fungal membrane stress might have an indirect cross-effect on susceptibility in *C. albicans*. This phenomenon needs further investigation in the future, as it might provide new information about the antifungal mechanism of NFAP2. The manuscript has been improved with the discussion of this reviewer's comment (L440-445).

In FIG. 1, there is no statistics analysis for data?

ANSWER: During the adaptation phase (FIG. 1) of the microevolution experiment, our goal was to generate six-six independently evolved cell lines. Our main aim with this experiment was to determine the concentration of the antifungal compound to which a certain cell line cannot adapt. All of the subjected strains behaved uniformly within their respective groups (NFAP2- or FLC-treated); therefore, we believe (a comprehensive) statistical analysis is not necessary.

Section of discussion is too long, and some are not important.

We would greatly appreciate it if the reviewer could disregard this comment, as we feel that the context of „Discussion” is complete in its current form. We think, all provided information is necessary to explain and understand the various behavior of different cell lines in the conducted experiments focusing on cell morphology, antifungal susceptibility, stress tolerance, metabolic fitness and virulence.

In virulence test, they stated that NFAP2 resistance does not enhance the virulence of C. albicans compared with FLC resistance.¹ However, in the curve of virulence, it does showing NFAP2 resistance affected virulence?¹ Many curves mixed together.²

ANSWERS:

¹: We would like to emphasize, that our findings indicate NFAP2 resistance (tolerance) does not influence virulence (NFAP2/4, NFAP2/5) or it diminishes virulence (NFAP2/6). This is supported by the statistical analysis. It concluded that the virulence of NFAP2/4 and NFAP2/5 is not significantly different compared to that of wild-type, while the virulence of NFAP2/6 is significantly lower. Therefore, based on these

statistics, we concluded that NFAP2 resistance (tolerance) does not enhance the virulence of *C. albicans*. Accordingly, our conclusion is right, NFAP2 resistance (tolerance) does not enhance the virulence. It is discussed in “Results” and “Discussion” respectively as the followings: L305-306: “*Thus, compared with FLC resistance, NFAP2 tolerance does not enhance the virulence of C. albicans.*” L458-459: “*However, the metabolic fitness of NFAP2-tolerant C. albicans remained unchanged and no increase in virulence was detected compared with FLC-resistant strains.*”

²: We thank the reviewer for the remark, in FIG. 5 we separated the virulence curves of the NFAP2-resistant (tolerant) and FLC-resistant strains into two panels for clarity, panel A is dedicated for NFAP2-resistant (tolerant) and panel B for FLC-resistant strains.

With kindest regards.

Yours sincerely,

László Galgóczy
associate professor
(on behalf of co-authors)

Re: Spectrum01273-24R1 (The *Neosartorya (Aspergillus) fischeri* antifungal protein NFAP2 has low potential to trigger resistance development in *Candida albicans in vitro*)

Dear Dr. László Galgóczy:

Your manuscript is now ready for publication. Congratulations !!!!!

Your manuscript has been accepted, and I am forwarding it to the ASM production staff for publication. Your paper will first be checked to make sure all elements meet the technical requirements. ASM staff will contact you if anything needs to be revised before copyediting and production can begin. Otherwise, you will be notified when your proofs are ready to be viewed.

Sincerely,
Gustavo Goldman
Editor
Microbiology Spectrum

Reviewer #1 (Comments for the Author):

The authors have answered all my queries and questions - I am happy with the resulting manuscript.

Reviewer #2 (Comments for the Author):

I think authors have response my all concerns and I have no more comments.